# Mega-scale movie-fields in the mouse visuo-hippocampal network

Chinmay Purandare[1,2,3]*[†], Mayank Mehta[2,3,4]*

[1]Department of Bioengineering, University of California, Los Angeles, Los Angeles, United States; [2]W.M. Keck Center for Neurophysics, Department of Physics and Astronomy, University of California, Los Angeles, Los Angeles, United States; [3]Department of Neurology, University of California, Los Angeles, Los Angeles, United States; [4]Department of Electrical and Computer Engineering, University of California, Los Angeles, Los Angeles, United States

**\*For correspondence:**
chinmay.purandare@gmail.com (CP);
MayankMehta@ucla.edu (MM)

**Present address:** [†]Department of Physiology, UCSF, San Francisco, United States

**Competing interest:** The authors declare that no competing interests exist.

**Abstract** Natural visual experience involves a continuous series of related images while the subject is immobile. How does the cortico-hippocampal circuit process a visual episode? The hippocampus is crucial for episodic memory, but most rodent single unit studies require spatial exploration or active engagement. Hence, we investigated neural responses to a silent movie (Allen Brain Observatory) in head-fixed mice without any task or locomotion demands, or rewards. Surprisingly, a third (33%, 3379/10263) of hippocampal –dentate gyrus, CA3, CA1 and subiculum– neurons showed movie-selectivity, with elevated firing in specific movie sub-segments, termed movie-fields, similar to the vast majority of thalamo-cortical (LGN, V1, AM-PM) neurons (97%, 6554/6785). Movie-tuning remained intact in immobile or spontaneously running mice. Visual neurons had >5 movie-fields per cell, but only ~2 in hippocampus. The movie-field durations in all brain regions spanned an unprecedented 1000-fold range: from 0.02s to 20s, termed mega-scale coding. Yet, the total duration of all the movie-fields of a cell was comparable across neurons and brain regions. The hippocampal responses thus showed greater continuous-sequence encoding than visual areas, as evidenced by fewer and broader movie-fields than in visual areas. Consistently, repeated presentation of the movie images in a fixed, but scrambled sequence virtually abolished hippocampal but not visual-cortical selectivity. The preference for continuous, compared to scrambled sequence was eight-fold greater in hippocampal than visual areas, further supporting episodic-sequence encoding. Movies could thus provide a unified way to probe neural mechanisms of episodic information processing and memory, even in immobile subjects, across brain regions, and species.

## eLife assessment

This manuscript analyzes large-scale Neuropixels recordings from visual areas and hippocampus of mice passively viewing repeated clips of a movie and reports that neurons respond with elevated firing activities to specific, continuous sequences of movie frames. The **important** results support a role of rodent hippocampal neurons in general episode encoding and advance understanding of visual information processing across different brain regions. The strength of evidence for the primary conclusion was found to be **convincing**.

## Introduction

In addition to the position and orientation of simple visual cues, like Gabor patches and drifting gratings (*Hubel and Wiesel, 1959*), primary visual cortical responses are also direction selective (*De Valois et al., 1982*), and show predictive coding (*Xu et al., 2012*), suggesting that the temporal sequence of

visual cues influences neural firing. Accordingly, these as well as higher visual cortical neurons encode a sequence of visual images, i.e., a movie (*de Vries et al., 2020*; *Yen et al., 2007*; *Herikstad et al., 2011*; *Vinje and Gallant, 2000*; *Froudarakis et al., 2014*; *Hoseini et al., 2019*; *Herikstad et al., 2011*; *Kampa et al., 2011*). The hippocampus is farthest downstream from the retina in the visual circuit. The rodent hippocampal place cells encode spatial or temporal sequences (*MacDonald et al., 2011*; *Mehta et al., 2000*; *Mehta and Wilson, 2000*; *Mehta, 2015*; *Mehta et al., 1997*; *Buzsáki and Moser, 2013*; *Mau et al., 2018*; *Kraus et al., 2015*; *Kraus et al., 2013*; *O'Keefe and Nadel, 1978*) and episode-like responses (*Pastalkova et al., 2008*; *Moore et al., 2021*; *Buzsáki and Tingley, 2018*). However, these responses typically require active locomotion (*McNaughton et al., 1996*), and they are thought to be non-sensory responses (*O'Keefe and Dostrovsky, 1971*). Primate and human hippocampal responses are selective to specific sets of visual cues, e.g., the objectplace association (*Parkinson et al., 1988*), their short-term (*Scoville and Milber, 1957*) and long-term (*Quiroga et al., 2005*) memories, cognitive boundaries between episodic movies (*Zheng et al., 2022*), and event integration for narrative association (*Cohn-Sheehy et al., 2021*). However, despite strong evidence for the role of hippocampus in episodic memory, the hippocampal encoding of a continuous sequence of images, i.e., a visual episode, is unknown.

## Results

### Significant movie tuning across cortico-hippocampal areas

We used a publicly available dataset (Allen Brain Observatory – Neuropixels Visual Coding, 2019 Allen Institute). Mice were monocularly shown a 30-s clip of a continuous segment from the movie *Touch of Evil* (Welles, 1958) (*Siegle et al., 2021*; *Figure 1—figure supplement 1* and *Figure 1—video 1*). Mice were head-fixed but were free to run on a circular disk. A total of 17,048 broad spiking, active, putatively excitatory neurons were analyzed, recorded using 4–6 Neuropixel probes in 24 sessions from 24 mice (see *Methods*).

The majority of neurons in the visual areas (lateral geniculate nucleus [LGN], primary visual cortex [V1], higher visual areas: antero-medial and posterior-medial [AM–PM]) were modulated by the movie, consistent with previous reports (*Figure 1—figure supplement 2*; *de Vries et al., 2020*; *Yen et al., 2007*; *Herikstad et al., 2011*; *Vinje and Gallant, 2000*; *Froudarakis et al., 2014*; *Hoseini et al., 2019*; *Kampa et al., 2011*). Surprisingly, neurons from all parts of the hippocampus (dentate gyrus [DG], CA3, CA1, subiculum [SUB]) were also clearly modulated (*Figure 1*), with reliable, elevated spiking across many trials in small movie segments. To quantify selectivity in an identical, firing rate- and threshold-independent fashion across brain regions, we computed the *z*-scored sparsity (*Acharya et al., 2016*; *Aghajan et al., 2015*; *Skaggs et al., 1996*; *Purandare et al., 2022*) of neural selectivity (see *Methods*). Cells with *z*-scored sparsity >2 were considered significantly (p < 0.03) modulated. Other metrics of selectivity, like depth of modulation or mutual information, provided qualitatively similar results (*Figure 1—figure supplement 3*). The areas V1 (97.3%) and AM–PM (97.1%) had the largest percentage of movie-tuned cells. Similarly, the majority of neurons in LGN (89.2%) too showed significant modulation by the movie. This level of selectivity is much higher than reported earlier (*de Vries et al., 2020*) (~40%), perhaps because we analyzed extracellular spikes, while the previous study used calcium imaging. On the other hand, the movie selectivity was greater than the selectivity for classic stimuli, like drifting gratings, in V1, even within calcium imaging data, in agreement of reports of better model fit with natural stimuli for primate visual responses (*David et al., 2004*). Direct quantitative comparison across stimuli is difficult and beyond the scope of this study because the movie frames appeared every 30 ms, and were preceded by similar images, while classic stimuli were presented for 250 ms, in a random order. Thus, the vast majority of thalamo-cortical neurons were significantly modulated by the movie.

Movie selectivity was prevalent in the hippocampal regions too, despite head fixation, dissociation between self-movements and visual cues as well as the absence of rewards, task, or memory demands (*Figure 1a–d*). Subiculum, the output region of the hippocampus, farthest removed from the retina, had the largest fraction (44.6%, *Figure 1d*) of movie-tuned neurons, followed by the upstream CA1 (33.6%, *Figure 1c*) and DG (33.1%, *Figure 1a*). However, CA3 movie selectivity was nearly half as much (17.3%, *Figure 1b*). This is unlike place cells, where CA3 and CA1 selectivity are comparable

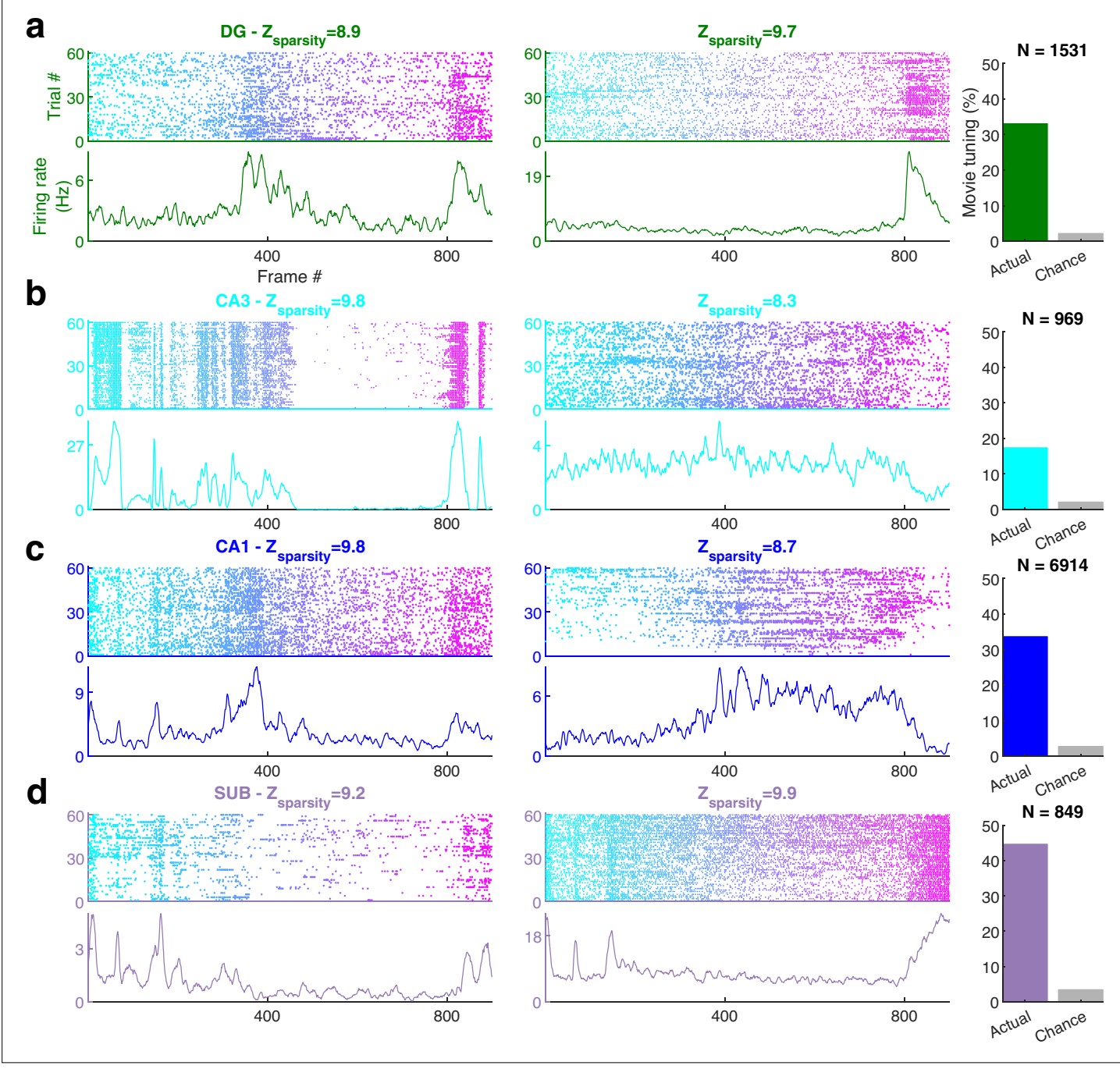

**Figure 1.** Movie frame selectivity in hippocampal neurons. (**a**) Raster plots of two different dentate gyrus (DG) neurons as a function of the movie frame (top) over 60 trials, and the corresponding mean firing rate response (bottom). These two cells had significantly increased activity in specific segments of the movie. Z-scored sparsity indicating strength of modulation is indicated above. 33.1% of dentate neurons were significantly modulated by the movie (right, green bar), far greater than chance (gray bar). Total active, broad spiking neurons for each brain region indicated at top ($N_{tuned}/N_{cells} = 506/1531$). (**b**) Same as (**a**), for CA3 (168/969, 17.3%), (**c**) CA1 (2326/6914, 33.6%), and (**d**) subiculum (379/849, 44.6%) neurons.

The online version of this article includes the following video and figure supplement(s) for figure 1:

**Figure supplement 1.** The movie.

**Figure supplement 2.** Movie selectivity across brain areas.

**Figure supplement 3.** Multiple metrics show significant and comparable movie tuning.

**Figure supplement 4.** Movie tuning is intact during immobility.

*Figure 1 continued on next page*

*Figure 1 continued*

**Figure supplement 5.** Simultaneously recorded hippocampal cells have different movie tuning.

**Figure supplement 6.** Movie tuning in unaffected by the removal of sharp-wave ripple (SWR) events.

**Figure supplement 7.** Movie tuning is comparable across sessions with or without prolonged stationary behavior, high or low pupil dilation or theta power.

**Figure supplement 8.** Movie presentation did not alter hippocampal firing rates and the mega-scale coding was unrelated to cluster quality.

**Figure 1—video 1.** Sequential movie.

https://elifesciences.org/articles/85069/figures#fig1video1

(*Jung and McNaughton, 1993*; *Muller, 1996*) and subiculum selectivity is weaker (*Sharp and Green, 1994*).

## Movie tuning is not an artifact of behavioral or brain state changes

To confirm these findings, we performed several controls. Running alters neural activity in visual areas (*Niell and Stryker, 2010*; *Erisken et al., 2014*; *Christensen and Pillow, 2022*; *Lee et al., 2014*) and hippocampus (*Góis and Tort, 2018*; *Wiener et al., 1989*; *Shan et al., 2016*). Hence, we used the data from only the stationary epochs (see *Methods*) and only from sessions with at least 300 s of stationary data (17 sessions, 24,906 cells). Movie tuning was unchanged in these data (*Figure 1—figure supplement 4*). This is unlike place cells where spatial selectivity is greatly reduced during immobility (*Chen et al., 2013*; *Foster et al., 1989*). Neurons recorded simultaneously from the same brain region also showed different selectivity patterns (*Figure 1—figure supplement 5*). Thus, nonspecific effects such as running cannot explain brain-wide movie selectivity. Prolonged immobility could change the brain state, e.g., the emergence of sharp-wave ripples (SWRs). Hence, we removed the data around SWRs and confirmed that movie tuning was unaffected (*Figure 1—figure supplement 6*). Strong movie-tuned cells were seen in sessions with long bouts of running as well as with predominantly immobile behavior (*Figure 1—figure supplement 7*), unlike responses to auditory tones, which were lost during running behavior (*Shan et al., 2016*). Place cell selectivity of hippocampal neurons is influenced by theta rhythm (*Foster and Wilson, 2007*; *Royer et al., 2012*; *Huxter et al., 2008*). We compared the movie selectivity during periods of high theta, versus periods of low theta. Significant movie selectivity in both cases (*Figure 1—figure supplement 7*). To further assess the effect of changes in brain state, we similarly analyzed movie tuning in two equal subsegments of data, corresponding to epochs with high and low pupil dilation, which is a strong correlate of arousal (*Vinck et al., 2015*; *Schröder et al., 2020*; *Fekete et al., 2009*). Movie tuning was above chance levels in both subsegments (*Figure 1—figure supplement 7*). Hence, locomotion, arousal, or changes in brain states cannot explain the hippocampal movie tuning.

## Similarities and differences between place-fields and movie-fields

Hippocampal neurons have one or two place-fields in typical mazes which take a few seconds to traverse (*O'Keefe and Burgess, 1996*). In larger arenas that take tens of seconds to traverse, the number of peaks per cell and the peak duration increases (*Eliav et al., 2021*; *Kjelstrup et al., 2008*; *Harland et al., 2021*; *Rich et al., 2014*). Peak detection for movie tuning is nontrivial because neurons have nonzero background firing rates, and the elevated rates cover a wide range (*Figure 1*). We developed a novel algorithm to address this (see *Methods*). On average, V1 neurons had the largest number of movie-fields (*Figure 2a*, mean ± standard error of the mean [SEM] = 10.4 ± 0.1, here we use mean instead of median to gain a better resolution for the small and discrete values of number of fields per cell), followed by LGN (8.6 ± 0.3) and AM–PM (6.3 ± 0.07). Hippocampal areas had significantly fewer movie-fields per cell: DG (2.1 ± 0.1), CA3 (2.8 ± 0.3), CA1 (2.0 ± 0.02), and subiculum (2.1 ± 0.05). Thus, the number of movie-fields per cell was smaller than the number of place-fields per cell in comparably long spatial tracks (*Eliav et al., 2021*; *Kjelstrup et al., 2008*; *Harland et al., 2021*; *Rich et al., 2014*; *Fenton et al., 2008*; *Park et al., 2011*), but a handful of hippocampal cells had more than five movie-fields (*Figure 2—figure supplement 1*).

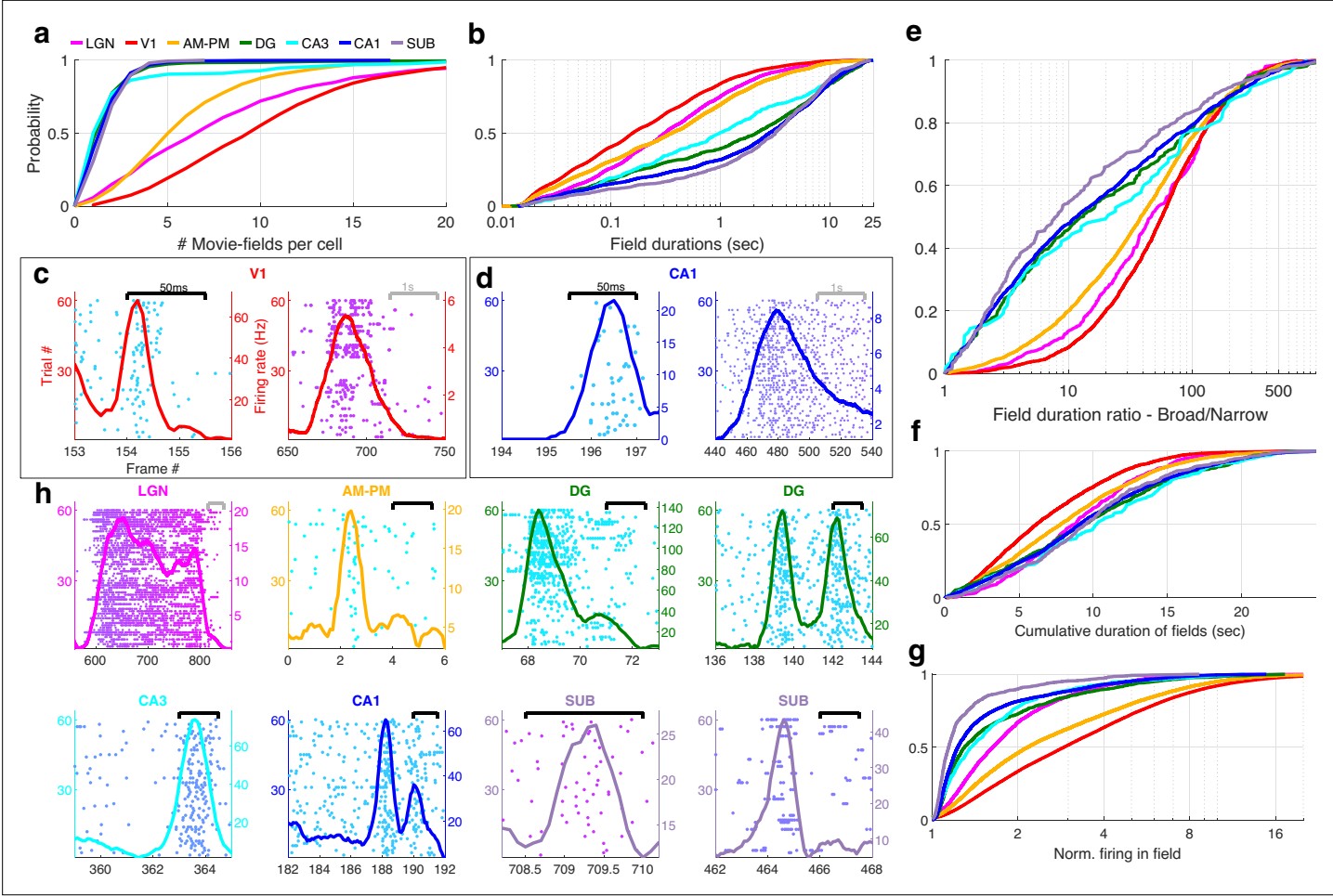

**Figure 2.** Multi-peaked, mega-scale movie-fields across all brain areas. (**a**) Distribution of the number of movie-fields per tuned cell (see *Methods*) in different brain regions (shown by different colors, top line inset, arranged in their hierarchical order). Hippocampal regions (blue-green shades) were significantly different from each other (KS-test p < 0.04), except DG–CA3. All visual regions were significantly different from each other (KS-test p < 7.0 × 10⁻¹¹). All visual–hippocampal region pairwise comparisons were also significantly different (KS-test p < 1.8 × 10⁻⁴⁴). CA1 had the lowest number of movie-fields per cell (2.0 ± 0.02, mean ± standard error of the mean [SEM]) while V1 had the highest (10.4 ± 0.1). (**b**) Distribution of the durations of movie-fields identified in (**a**), across all tuned neurons from a given brain region. These were significantly different for all brain region pairs (KS-test p < 7.3 × 10⁻³). The longest movie-fields were in subiculum (median ± SEM, here and subsequently, unless otherwise mentioned, 3169.9 ± 169.8 ms), and the shortest in V1 (156.6 ± 9.2 ms). (**c**) Snippets of movie-fields from an example cell from V1, with two of the fields zoomed in, showing 60× difference in duration. Black bar at top indicates 50 ms, and gray bar indicates 1 s. Each frame corresponds to 33.3 ms. Average response (solid trace, *y*-axis on the right) is superimposed on the trial wise spiking response (dots, *y*-axis on the left). Color of dots corresponds to frame numbers as in *Figure 1*. (**d**) Same as (**c**), for a CA1 neuron with 54× difference in duration. (**e**) The ratio of longest to shortest field duration within a single cell, i.e., mega-scale index, was largest in V1 (56.7 ± 2.2) and least in subiculum (8.0 ± 9.7). All visual–visual and visual–hippocampal brain region pairs were significantly different on this metric (KS-test p < 0.02). Among the hippocampal–hippocampal pairs, only CA3–SUB were significantly different (p = 0.03). (**f**) For each cell, the total duration of all movie-fields, i.e., cumulative duration of significantly elevated activity, was comparable across brain regions. The largest cumulative duration (10.2 ± 0.46 s, CA3) was only 1.66× of the smallest (6.2 ± 0.09 s) (V1). Visual–hippocampal and visual–visual brain region pairs' cumulative duration distributions were significantly different (KS-test p < 0.001), but not hippocampal pairs (p > 0.07). (**g**) Distribution of the firing within fields, normalized by that in the shuffle response. All fields from all tuned neurons in a brain region were used. Firing in movie-fields was significantly different across all brain region pairs (KS-test, p < 1.0 × 10⁻⁷), except DG–CA3. Movie-field firing was largest in V1 (2.9 ± 0.03) and smallest in subiculum (1.14 ± 0.03). (**h**) Snippets of movie-fields from representative tuned cells, from lateral geniculate nucleus (LGN) showing a long movie-field (233 frames, or 7.8 s, panel 1), and from AM–PM and from hippocampus showing short fields (two frames or 66.6 ms wide or less).

The online version of this article includes the following figure supplement(s) for figure 2:

**Figure supplement 1.** Few hippocampal neurons had greater than five movie-fields.

**Figure supplement 2.** Mega-scale movie-coding within a single cell is smaller than the ensemble wide mega-scale index in visual, but not hippocampal areas.

## Mega-scale structure of movie-fields

Typical receptive field size increases as one moves away from the retina in the visual hierarchy (*Siegle et al., 2021*). A similar effect was seen for movie-field durations. On average, hippocampal movie-fields were longer than visual regions (*Figure 2b*). But there were many exceptions –movie-fields of LGN (median ± SEM, here and subsequently, unless stated otherwise, 308.5 ± 33.9 ms) were twice as long as in V1 (156.6 ± 9.2 ms). Movie-fields of subiculum (3169.9 ± 169.8 ms) were significantly longer than CA1 (2786.1 ± 77.5 ms) and nearly threefold longer than the upstream CA3 (979.1 ± 241.1 ms). However, the dentate movie-fields (2113.2 ± 172.4 ms) were twofold longer than the downstream CA3. This is similar to the patterns reported for CA3, CA1, and DG place cells (*Park et al., 2011*). But others have claimed that CA3 place-fields are slightly bigger than CA1 (*Roth et al., 2012*), whereas movie-fields showed the opposite pattern.

The movie-field durations spanned a 500- to 1000-fold range in every brain region investigated (*Figure 2e*). This mega-scale scale is unprecedentedly large, nearly two orders of magnitude greater than previous reports in place cells (*Eliav et al., 2021*; *Harland et al., 2021*). Even individual neurons showed 100-fold mega-scale responses (*Figure 2c, d*) compared to less than 10-fold scale within single place cells (*Eliav et al., 2021*; *Harland et al., 2021*). The mega-scale tuning within a neuron was largest in V1 and smallest in subiculum (*Figure 2e*). This is partly because the short-duration movie-fields in hippocampal regions were typically neither as narrow nor as prominent as in the visual areas (*Figure 2—figure supplement 2*).

Despite these differences in mega-scale tuning across different brain areas, the total duration of elevated activity, i.e., the cumulative sum of movie-field durations within a single cell, was remarkably conserved across neurons within and across brain regions (*Figure 2f*). Unlike movie-field durations, which differed by more than tenfold between hippocampal and visual regions, cumulative durations were quite comparable, ranging from 6.2 s (V1) to 10.2 s (CA3) (*Figure 2f*, LGN = 8.8 ± 0.21 s, V1 = 6.2 ± 0.09, AM–PM = 7.8 ± 0.09, DG = 9.4 ± 0.26, CA3 = 10.2 ± 0.46, CA1 = 9.1 ± 0.12, SUB = 9.5 ± 0.27). Thus, hippocampal movie-fields are longer and less multi-peaked than visual areas, such that the total duration of elevated activity was similar across all areas, spanning about a fourth of the movie, comparable to the fraction of large environments in which place cells are active (*Harland et al., 2021*; *Fenton et al., 2008*; *Park et al., 2011*). To quantify the net activity in the movie-fields, we computed the total firing in the movie-fields (i.e., the area under the curve for the duration of the movie-fields), normalized by the expected discharge from the shuffled response. Unlike the tenfold variation of movie-field durations, net movie-field discharge was more comparable (<3× variation) across brain areas, but maximal in V1 and least in subiculum (*Figure 2g*).

Many movie-fields showed elevated activity spanning up to several seconds, suggesting rate-code like encoding (*Figure 2h*). However, some cells showed movie-fields with elevated spiking restricted to less than 50 ms, similar to responses to briefly flashed stimuli in anesthetized cats (*Yen et al., 2007*; *Heriksted et al., 2011*; *Xia et al., 2021*). This is suggestive of a temporal code, characterized by low spike timing jitter (*Ikegaya et al., 2004*). Such short-duration movie-fields were not only common in the thalamus (LGN), but also AM–PM, three synapses away from the retina. A small fraction of cells in the hippocampal areas, more than five synapses away from the retina, showed such temporally coded fields as well (*Figure 2h*).

To determine the stability and temporal-continuity of movie tuning across the neural ensembles we computed the population vector overlap between even and odd trials (*Resnik et al., 2012*) (see *Methods*). Population response stability was significantly greater for tuned than for untuned neurons (*Figure 3—figure supplement 1*). The population vector overlap around the diagonal was broader in hippocampal regions than visual cortical and LGN, indicating longer temporal-continuity, reflective of their longer movie-fields. Furthermore, the population vector overlap away from the diagonal was larger around frames 400–800 in all brain areas due to the longer movie-fields in that movie segment (see below).

## Relationship between movie image content and neural movie tuning

Are all movie frames represented equally by all brain areas? The duration and density of movie-fields varied as a function of the movie frame and brain region (*Figure 3—figure supplement 2*). We hypothesized that this variation could correspond to the change in visual content from one frame to the next. Hence, we quantified the similarity between adjacent movie frames as the correlation

coefficient between corresponding pixels and termed it as frame-to-frame (F2F) image correlation. For comparison, we also quantified the similarity between the neural responses to adjacent frames (F2F neural correlation), as the correlation coefficient between the firing rate response of neuronal ensembles between adjacent frames. For all brain regions, the neural F2F was correlated with image F2F, but this correlation was weaker in hippocampal output regions (CA1 and SUB) than visual regions like LGN and V1. The majority of brain regions had substantially reduced density of movie-fields between the movie frames 400–800, but the movie-fields were longer in this region. This effect as well was greater in the visual regions than hippocampal regions. Using significantly tuned neurons, we computed the average neural activity in each brain region at each point in the movie (see *Methods*). Although movie-fields (*Figure 3a*), or just the strongest movie-field per cell (*Figure 3b*), covered the entire movie, the peak normalized, ensemble activity level of all brain regions showed significant overrepresentation, i.e., deviation from the uniformity, in certain parts of the movie (*Figure 3c*, see *Methods*). This was most pronounced in V1 and the higher visual areas AM–PM. The number of movie frames with elevated ensemble activity was higher in visual cortical areas than hippocampal regions (*Figure 3d*), and also this modulation (see *Methods*) was smaller in hippocampus and LGN, compared to the visual cortical regions (*Figure 3e*).

Using the significantly tuned neurons, we also computed the average neural activity in each brain region corresponding to each frame in the movie, without peak rate normalization (see *Methods*). The degree of continuity between the movie frames, quantified as above (F2F image correlation), was inversely correlated with the ensemble rate modulation in all areas except DG, CA3, and CA1 (*Figure 3f, g*). As expected for a continuous movie, this F2F image correlation was close to unity for most frames, but highest in the latter part of the movie where the images changed more slowly. The population wide elevated firing rates, as well as the smallest movie-fields, occurred during the earlier parts (*Figure 3—figure supplement 2*). Thus, the movie-code was stronger in the segments with greatest change across movie frames, in agreement with recent reports of visual cortical encoding of flow stimuli (*Dyballa et al., 2018*). These results show differential population representation of the movie across brain regions.

## Differential neural encoding of sequential versus scrambled movie in visual and hippocampal areas

If these responses were purely visual, a movie made of scrambled sequence of images would generate equally strong or even stronger selectivity due to the even larger change across movie frames, despite the absence of similarity between adjacent frames. To explore this possibility, we investigated neural selectivity when the same movie frames were presented in a fixed but scrambled sequence (scrambled movie, *Figure 4—video 1*). The within frame and the total visual content were identical between the continuous and scrambled movies, and the same sequence of images was repeated many times in both experiments (see *Methods*). But there was no correlation between adjacent frames, i.e., visual continuity, in the latter (*Figure 4a*).

For all brain regions investigated, the continuous movie generated significantly greater modulation of neural activity than the scrambled sequence (*Figure 4b*). Middle 20 trials of the continuous movie were chosen as the appropriate subset for comparison since they were chronologically closest to the scrambled movie presentation. This choice ensured that other long-term effects, such as behavioral state change, instability of single-unit measurement and representational (*Deitch et al., 2021*) or behavioral (*Sadeh and Clopath, 2022*) drift could not account for the differences in neural responses to continuous and scrambled movie presentation. This preference for continuous over scrambled movie was the greatest in hippocampal regions where the percentage of significantly tuned neurons (4.4%, near chance level of 2.3%) reduced more than fourfold compared to the continuous movie (17.8%, after accounting for the lesser number of trials, see *Methods*). This was unlike visual areas where the scrambled (80.4%) and the continuous movie (92.4%) generated similar prevalence levels of selectivity (*Figure 4b*). The few hippocampal cells which had significant selectivity to the scrambled sequence, did not have long-duration responses, but only very short, ~50-ms long responses (*Figure 4d*), reminiscent of, but even sharper than human hippocampal responses to flashed images (*Quiroga et al., 2005*). To estimate the effect of continuous movie compared to the scrambled sequence on individual cells, we computed the normalized difference between the continuous and scrambled movie selectivity for cells which were selective in either condition (*Figure 4c*, see *Methods*).

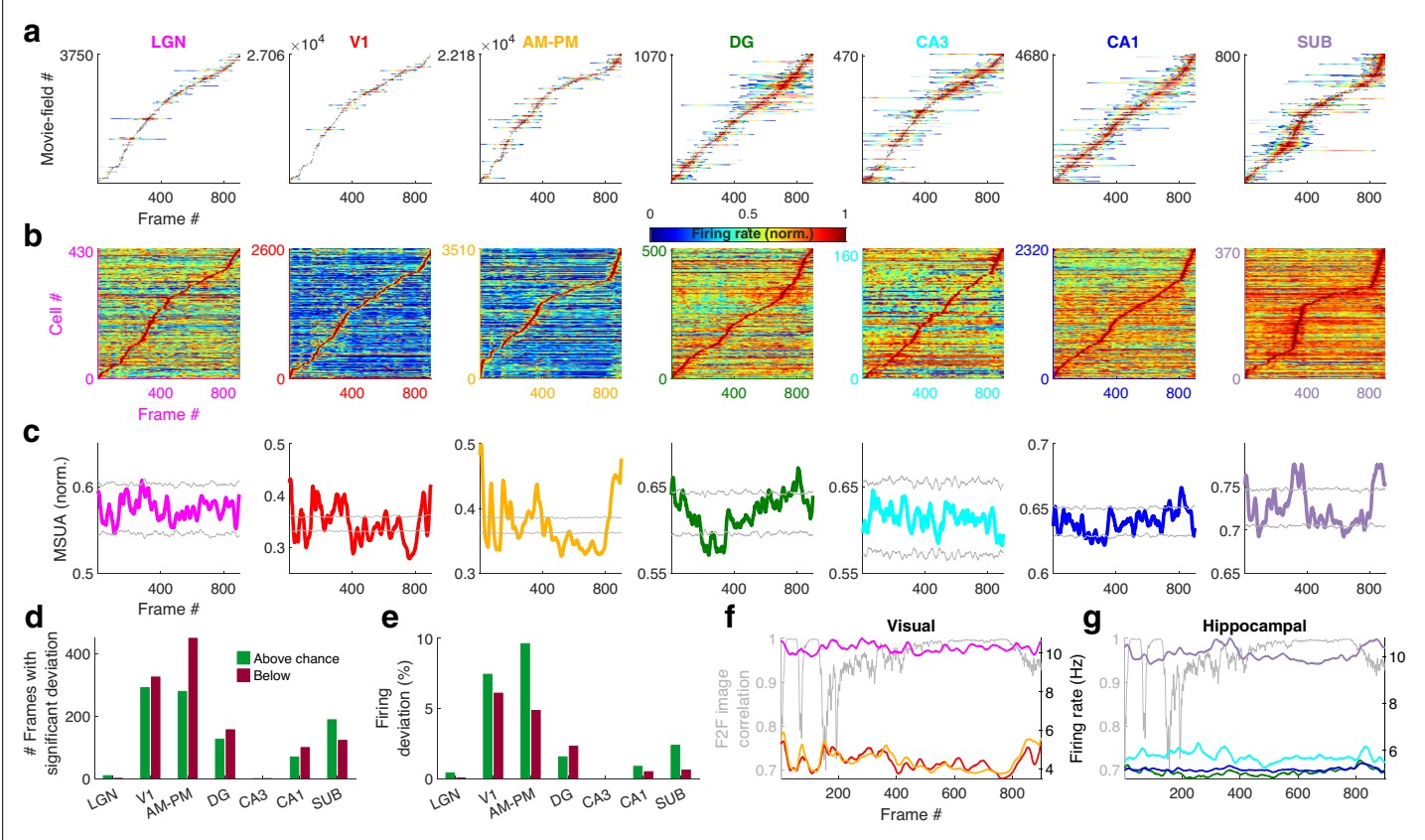

**Figure 3.** Population averaged movie tuning varies across brain areas. (**a**) Stack plot of all the movie-fields detected from all tuned neurons of a brain region. Color indicates relative firing rate, normalized by the maximum firing rate in that movie-field. The movie-fields were sorted according to the frame with the maximal response. Note accumulation of fields in certain parts of the movie, especially in subiculum and AM–PM. (**b**) Similar to (**a**), but using only a single, tallest movie-field peak from each neuron showing a similar pattern, with pronounced overrepresentation of some portions of the movie in most brain areas. Each neuron's response was normalized by its maximum firing rate. The average firing rate of non-peak frames, which was inversely related to the depth of modulation, was smallest (0.35× of the average peak response across all neurons) for V1, followed by AM–PM 0.37, leading to blue shades. Average non-peak responses were higher for other regions (0.57× the peak for LGN, CA3 – 0.61, DG – 0.62, CA1 – 0.64, and SUB – 0.76), leading to warmer off-diagonal colors. (**c**) Multiple single-unit activity (MSUA) in a given brain region, obtained as the average response across all tuned cells, by using maxima-normalized response for each cell from (**b**). Gray lines indicate mean ± 4*std response from the shuffle data corresponding to p = 0.025 after Bonferroni correction for multiple comparisons (see *Methods*). AM–PM had the largest MSUA modulation (sparsity = 0.01) and CA1 had the smallest (sparsity = $1.8 \times 10^{-4}$). The MSUA modulation across several brain region pairs – AM&PM–DG, V1–CA3, DG–CA3, CA3–CA1, and CA1–SUB were not significantly correlated (Pearson correlation coefficient p > 0.05). Some brain region pairs, DG–LGN, DG–V1, AM&PM–CA3, LGN–CA1, V1–CA1, DG–SUB, and CA3–SUB, were significantly negatively correlated ($r < -0.18$, $p < 4.0 \times 10^{-7}$). All other brain region pairs were significantly positively correlated ($r > 0.07$, $p < 0.03$). (**d**) Number of frames for which the observed MSUA deviates from the $z = \pm 4$ range from (**c**), termed significant deviation. V1 and AM–PM had the largest positive deviant frames (289), and CA3 had the least (zero). Unlike CA3, the low number of deviant frames for LGN could not be explained by sample size, because there were more tuned cells in LGN than SUB. (**e**) Firing in deviant frames above (or below) chance level, as a percentage of the average response. Above chance level deviation was greater or equal to that below, for all brain regions except DG, with the largest positive deviation in AM–PM (9.3%), largest negative deviation in V1 (6.0%), and least in CA3 (zero each). (**f**) Total firing rate response of visual regions across tuned neurons. All regions had significant negative correlation ($r < -0.39$, $p < 3.4 \times 10^{-34}$) between the ensemble response and the frame-to-frame (F2F) image correlation (gray line, y-axis on the left) across movie frames. (**g**) Similar to (**f**), for hippocampal regions. CA3 response were not significantly correlated with the F2F correlation, dentate gyrus ($r = 0.26$, $p = 4.0 \times 10^{-15}$) and CA1 ($r = 0.21$, $p = 1.5 \times 10^{-10}$) responses were positively correlated, and subiculum response was negatively correlated ($r = -0.44$, $p = 2.2 \times 10^{-43}$). Note the substantially higher mean firing rates of LGN in (**f**) and subiculum neurons in (**g**) (colored lines closer to the top) compared to other brain areas.

The online version of this article includes the following figure supplement(s) for figure 3:

**Figure supplement 1.** Population vector overlap is wider in hippocampus than visual areas.

**Figure supplement 2.** Movie-field properties strongly reflect the frame-to-frame correlation structure of the movie in the visual but not hippocampal areas.

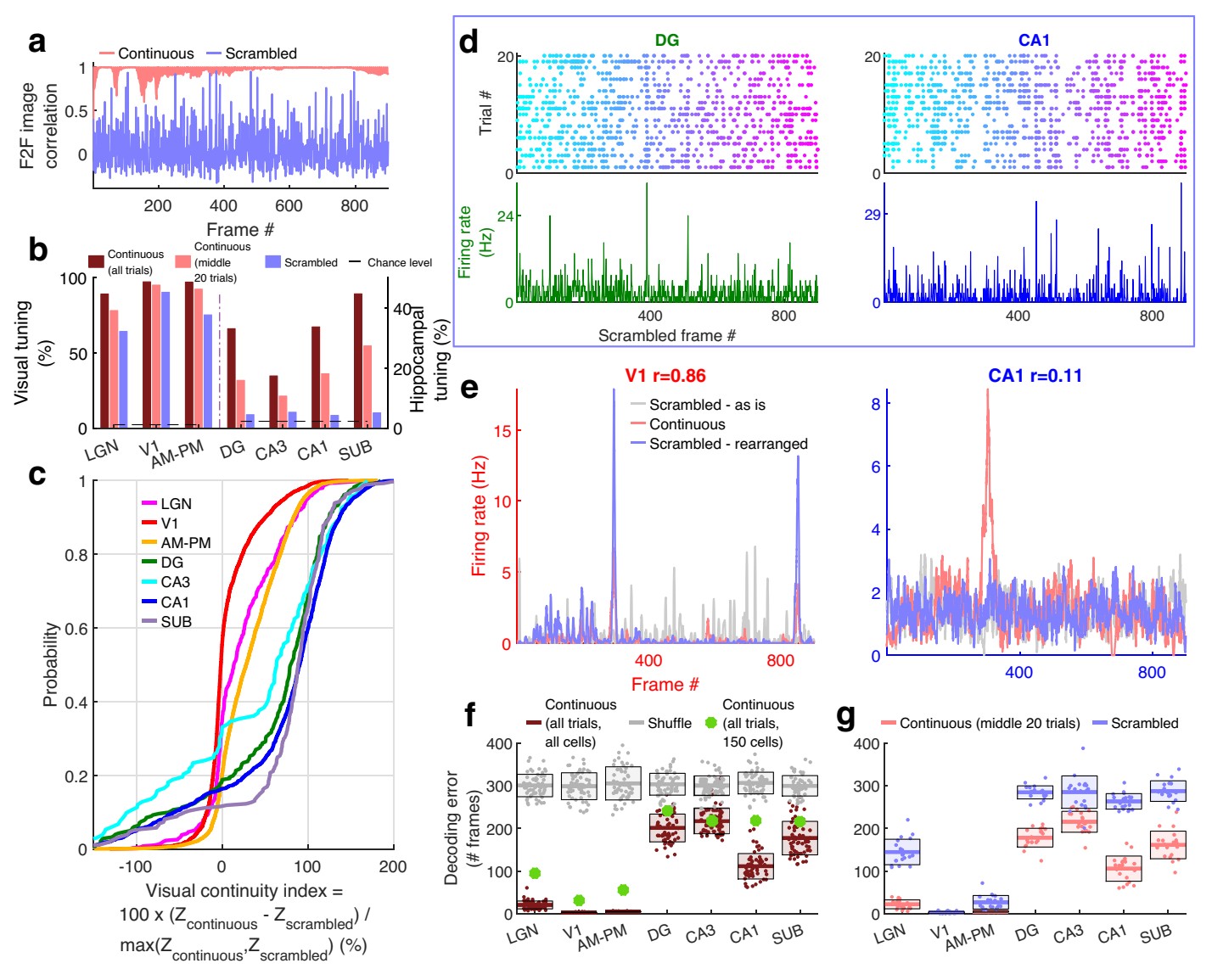

**Figure 4.** Larger reduction of selectivity in hippocampal than visual regions due to scrambled presentation. (**a**) Similarity between the visual content of one frame with the subsequent one, quantified as the *Pearson* correlation coefficient between pixel–pixel across adjacent frames for the continuous movie (pink) and the scrambled sequence (lavender), termed F2F image correlation. Similar to *Figure 3g*. For the scrambled movie, the frame number here corresponded to the chronological frame sequence, as presented. (**b**) Fraction of broad spiking neurons significantly modulated by the continuous movie (red) or the scrambled sequence (blue) using z-scored sparsity measures (similar to *Figure 1*, see *Methods*). For all brain regions, continuous movie generated greater selectivity than scrambled sequence (KS-test p < 7.4 × 10⁻⁴). (**c**) Percentage change in the magnitude of tuning between the continuous and scrambled movies for cells significantly modulated by either continuous or scrambled movie, termed visual continuity index. The largest drop in selectivity due to scrambled movie occurred in CA1 (90.3 ± 2.0%), and least in V1 (−1.5 ± 0.6%). Visual continuity index was significantly different between all brain region pairs (KS-test p < 0.03) and significantly greater for hippocampal areas than visual (8.2-fold, p < 10⁻¹⁰⁰). (**d**) Raster plots (top) and mean rate responses (color, bottom) showing increased spiking responses to only one or two scrambled movie frames, lasting about 50 ms. Tuned responses to scrambled movie were found in all brain regions, but these were the least frequent in DG and CA1. (**e**) One representative cell each from V1 (left) and CA1 (right), where the frame rearrangement of scrambled responses resulted in a response with high correlation to the continuous movie response for V1, but not CA1. Pearson correlation coefficient values of continuous movie and rearranged scrambled responses are indicated on top. (**f**) Average decoding error for observed data (see *Methods*), over 60 trials for continuous movie (maroon), was significantly lower than shuffled data (gray) (KS-test p < 1.2 × 10⁻²²). Solid line – mean error across 60 trials using all tuned cells from a brain region, shaded box – standard error of the mean (SEM), green dots – mean error across all trials using a random subsample of 150 cells from each brain region. Decoding error was lowest for V1 (30.9 frames) and highest in DG (241.2) and significantly different between all brain regions pairs (p < 1.9 × 10⁻⁴), except CA3–CA1, CA3–subiculum, and CA1–subiculum (p > 0.63). (**g**) Similar to (**f**), decoding of scrambled movie was significantly worse than that for the continuous movie (KS-test p < 2.6 × 10⁻³). Scrambled responses, in their 'as is', chronological order were used herein. Lateral geniculate nucleus (LGN) decoding error for scrambled presentation

*Figure 4 continued on next page*

*Figure 4 continued*

was 6.5× greater than that for continuous movie, whereas the difference in errors was least for V1 (1.04×). Scrambled movie decoding error for all visual areas and for CA1 and subiculum was significantly smaller than chance level (KS-test p < 2.6 × 10⁻³), but not DG and CA3 (p > 0.13). The middle 20 trials of the continuous movie were used for comparison with the scrambled movie since the scrambled movie was only presented 20 times. Middle trials of the continuous movie were chosen as the appropriate subset since they were chronologically closest to the scrambled movie presentation.

The online version of this article includes the following video and figure supplement(s) for figure 4:

**Figure supplement 1.** Scrambled movie elicits narrower but more movie-fields per cell than the continuous movie in all the visual regions.

**Figure supplement 2.** Cell by cell comparison of continuous versus scrambled movie responses.

**Figure supplement 3.** Multiple single-unit activity (MSUA) across all movie-tuned neurons in a brain region shows greater modulation than chance for the scrambled sequence in all visual areas.

**Figure supplement 4.** Latency of responses to the scrambled-sequence corresponds to the anatomical hierarchy of visual areas.

**Figure supplement 5.** Movie tuning in hippocampal neurons remains near chance level even after rearranging scrambled movie frames.

**Figure supplement 6.** Population vector overlap was narrower for the scrambled compared to the continuous movie.

**Figure 4—video 1.** Scrambled movie.

https://elifesciences.org/articles/85069/figures#fig4video1

This visual continuity index was more than eightfold higher in hippocampal areas (median values across all four hippocampal regions = 87.8%) compared to the visual areas (median = 10.6% across visual regions).

The pattern of increasing visual continuity index as we moved up the visual hierarchy, largely paralleled the anatomic organization (*Felleman and Van Essen, 1991*), with the greatest sensitivity to visual continuity in the hippocampal output regions, CA1 and subiculum, but there were notable exceptions. The primary visual cortical neurons showed the least reduction in selectivity due to the loss of temporally contiguous content, whereas LGN neurons, the primary source of input to the visual cortex and closer to the periphery, showed far greater sensitivity (*Figure 4c*).

Many visual cortical neurons were significantly modulated by the scrambled sequence, but their number of movie-fields per cell was greater and their duration was shorter than during the continuous movie (*Figure 4—figure supplements 1 and 2*). This could occur due to the loss of F2F correlation in the scrambled sequence. The average activity of the neural population in V1 and AM–PM showed significant deviation even with the scrambled movie, comparable to the continuous movie, but this multi-unit ensemble response was uncorrelated with the F2F correlation in the scrambled sequence (*Figure 4—figure supplement 3*). A substantial fraction of visual cortical and LGN responses to the scrambled sequence could be rearranged to resemble continuous movie responses (*Figure 4—figure supplement 4*, see *Methods*). The latency needed to shift the responses was least in LGN and largest in AM–PM, as expected from the feed-forward anatomy of visual information processing (*Siegle et al., 2021*; *Felleman and Van Essen, 1991*; *Figure 4—figure supplement 4*). Unlike visual areas, such rearrangement did not resemble the continuous movie responses in the hippocampal regions (example cells in *Figure 4e*, also see *Figure 4—figure supplement 4* for statistics and details). Furthermore, even after rearranging the hippocampal responses, their selectivity to the scrambled movie presentation remained near chance levels (*Figure 4—figure supplement 5*).

Population vector decoding of the ensemble of a few hundred place cells is sufficient to decode the rat's position using place cells (*Wilson and McNaughton, 1993*), and the position of a passively moving object (*Purandare et al., 2022*). Using similar methods, we decoded the movie frame number (see *Methods*). Continuous movie decoding was better than chance in all brain regions analyzed (*Figure 4f*). Upon accounting for the number of tuned neurons from different brain regions, the decoding was most accurate in V1, and least in DG. Scrambled movie decoding was significantly weaker yet above chance level (based on shuffles, see *Methods*) in visual areas, but not in CA3 and DG. But CA1 and subiculum neuronal ensembles could be used to decode scrambled movie frame number slightly above chance levels (*Figure 4g*). Similarly, the population overlap between even and odd trials for the scrambled sequence was strong for visual areas, and weaker in hippocampal regions, but significantly greater than untuned neurons in hippocampal regions (*Figure 4—figure supplement 6*). Combined with the handful of neurons in hippocampus whose movie selectivity persisted to the scrambled presentation, this suggests that loss of correlations between adjacent frames in the scrambled sequence abolishes most, but not all of the hippocampal selectivity to visual sequences.

## Discussion

### Movie tuning in the visual areas

To understand how neurons encode a continuously unfolding visual episode, we investigated the neural responses in the head-fixed mouse brain to an isoluminant, black-and-white, silent human movie, without any task demands or rewards. As expected, neural activity showed significant modulation in all thalamo-cortical visual areas, with elevated activity in response to specific parts of the movie, termed movie-fields. Most (96.6%, 6554/6785) of thalamo-cortical neurons showed significant movie tuning. This is nearly double that reported for the classic stimuli such as Gabor patches in the same dataset (*Siegle et al., 2021*), although a direct comparison is difficult due to the differences in experimental and analysis methods. For example, the classic stimuli were presented for 250 ms, preceded by a blank background whereas the images changed every 30 ms in a movie. On the other hand, significant tuning of the vast majority of visual neurons to movies is consistent with other reports (*de Vries et al., 2020*; *Yen et al., 2007*; *Herikstad et al., 2011*; *Froudarakis et al., 2014*; *Xia et al., 2021*; *Dyballa et al., 2018*; *Deitch et al., 2021*; *Sadeh and Clopath, 2022*). Thus, movies are a reliable method to probe the function of the visual brain and its role in cognition.

### Movie tuning in hippocampal areas

Remarkably, a third of hippocampal neurons (32.9%, 3379/10,263) were also movie tuned, comparable to the fraction of neurons with significant spatial selectivity in mice (*Jun et al., 2020*) and bats (*Yartsev et al., 2011*), and far greater than significant place cells in the primate hippocampus (*Rolls and O'Mara, 1995*; *Rolls, 2023*; *Mao et al., 2021*). While the hippocampus is implicated in episodic memory (*Vargha-Khadem et al., 1997*), rodent hippocampal responses are largely studied in the context of spatial maps or place cells (*O'Keefe and Nadel, 1978*) , and more recently in other tasks which requires active locomotion or active engagement (*Aronov et al., 2017*; *Danjo et al., 2018*). However, unlike place cells (*Chen et al., 2013*; *Foster et al., 1989*), movie tuning remained intact during immobility in all brain areas studied, which could be because self-motion causes consistent changes in multisensory cues during spatial exploration but not during movie presentation. This dissociation of the effect of mobility on spatial and movie selectivity agrees with the recent reports of dissociated mechanisms of episodic encoding and spatial navigation in human amnesia (*McAvan et al., 2022*). Our results are broadly consistent with prior studies that found movie selectivity in human hippocampal single neurons (*Gelbard-Sagiv et al., 2008*). However, that study relied on famous, very familiar movie clips, similar to the highly familiar image selectivity (*Quiroga et al., 2005*) to probe episodic memory recall. In contrast, mice in our study had seen this black-and-white, human movie clip only in two prior habituation sessions and it is very unlikely that they understood the episodic content of the movie. Recent studies found human hippocampal activation in response to abrupt changes between different movie clips (*Zheng et al., 2022*; *Cohn-Sheehy et al., 2021*; *Reagh and Ranganath, 2023*), which is broadly consistent with our findings. Future studies can investigate the nature of hippocampal activation in mice in response to familiar movies to probe episodic memory and recall. These observations support the hypothesis that specific visual cues can create reliable representations in all parts of hippocampus in rodents (*Chen et al., 2013*; *Acharya et al., 2016*; *Purandare et al., 2022*), nonhuman primates (*Rolls and O'Mara, 1995*; *Mao et al., 2021*), and humans (*Jacobs et al., 2010*; *Ekstrom et al., 2003*), unlike spatial selectivity which requires consistent information from multisensory cues (*Moore et al., 2021*; *Aghajan et al., 2015*; *Ravassard et al., 2013*).

### Mega-scale nature of movie-fields

Across all brain regions, neurons showed a mega-scale encoding by movie-fields varying in duration by up to 1000-fold, similar to, but far greater than recent reports of 10-fold multi-scale responses in the hippocampus (*Eliav et al., 2021*; *Kjelstrup et al., 2008*; *Harland et al., 2021*; *Rich et al., 2014*; *Fenton et al., 2008*; *Park et al., 2011*; *Harland et al., 2018*). While neural selectivity to movies has been studied in visual areas, such mega-scale coding has not been reported. Remarkably, mega-scale movie-coding was found not only across the population but even individual LGN and V1 neurons could show two different movie-fields, one lasting less than 100 ms and other exceeding 10,000 ms. The speed at which visual content changed across movie frames could explain a part, but not all of this effect. The mechanisms governing the mega-scale encoding would require additional studies. For example, the average duration of the movie-field increased along the feed-forward hierarchy,

consistent with the hierarchy of response lags during language processing (*Chang et al., 2022*). Paradoxically, the mega-scale coding of movie-field meant the opposite pattern also existed, with 10-s long movie-fields in some LGN cells while less than 100 ms long movie-fields in subiculum.

## Continuous versus scrambled movie responses

The analysis of scrambled movie-sequence allowed us to compute the neural response latency to movie frames. This was highest in AM–PM (91 ms) than V1 (74 ms) and least in LGN (60 ms), thus following the visual hierarchy. The pattern of movie tuning properties was also broadly consistent between V1 and AM/PM (*Figure 2*). However, several aspects of movie tuning did not follow the feed-forward anatomical hierarchy. For example, all metrics of movie selectivity (*Figure 2*) to the continuous movie showed a pattern that was the inconsistent to the feed-forward anatomical hierarchy: V1 had stronger movie tuning, higher number of movie-fields per cell, narrower movie-field widths, larger mega-scale structure, and better decoding than LGN. V1 was also more robust to scrambled sequence than LGN. One possible explanation is that there are other sources of inputs to V1, beyond LGN, that contribute significantly to movie tuning (*Spacek et al., 2022*). Among the hippocampal regions, the tuning properties of CA3 neurons (field durations, mega-chronicity index, visual continuity index, and several measures of population modulation) were closest to that of visual regions, even though the prevalence of tuning in CA3 was lesser than that in other hippocampal as well as visual areas.

## Emergence of episode-like movie code in hippocampus

Temporal integration window (*Norman-Haignere et al., 2022*; *Gauthier et al., 2012*; *Hasson et al., 2008*) as well as intrinsic timescale of firing (*Siegle et al., 2021*) increase along the anatomical hierarchy in the cortex, with the hippocampus being farthest removed from the retina (*Felleman and Van Essen, 1991*). This hierarchical anatomical organization, with visual areas being upstream of hippocampus could explain the longer movie-fields, the strength of tuning, number of movie peaks, their width, and decoding accuracy in hippocampal regions. This could also explain the several fold greater preference for the continuous movie over scrambled sequence in the hippocampus compared to the upstream visual areas. But, unlike reports of image-association memory in the inferior temporal cortex for unrelated images (*Sakai and Miyashita, 1991*; *Miyashita, 1988*), only a handful hippocampal neurons showed selective responses to the scrambled sequence. These results, along with the longer duration of hippocampal movie-fields could mediate visual-chunking or binding of a sequence of events. In fact, evidence for episodic-like chunking of visual information was found in all visual areas as well, where the scrambled-sequence not only reduced neural selectivity but caused fragmentation of movie-fields (*Figure 4—figure supplement 4*).

## No evidence of nonspecific effects

Could the brain-wide mega-scale tuning be an artifact of poor unit isolation, e.g., due to an erroneous mixing of two neurons, one with very short and another with very long movie-fields? This is unlikely since the LGN and visual cortical neural selectivity to classic stimuli (Gabor patches, drifting gratings, etc.) in the same dataset was similar to that reported in most studies (*Siegle et al., 2021*) whereas poor unit isolation should reduce these selective responses. However, to directly test this possibility, we calculated the correlation between the unit isolation index (or fraction of refractory violations) and the mega-scale index of the cell, while factoring out the contribution of mean firing rate (*Figure 1—figure supplement 8*). This correlation was not significant (p > 0.05) for any brain areas.

## Movie-fields versus place-fields

Do the movie-fields arise from the same mechanism as place-fields? Studies have shown that when rodents are passively moved along a linear track that they had explored (*Foster et al., 1989*), or when the images of the environment around a linear track was played back to them (*Chen et al., 2013*), some hippocampal neurons generated spatially selective activity. Since the movie clip involved change of spatial view, one could hypothesize that the movie-fields are just place-fields generated by passive viewing. This is unlikely for several reasons. Mega-scale movie-fields were found in the vast majority of all visual areas including LGN, far greater than spatially modulated neurons in the visual cortex during virtual navigation (*Haggerty and Ji, 2015*; *Saleem et al., 2018*). Furthermore, in prior passive viewing experiments, the rodents were shown the same narrow linear track, like a tunnel, that

they had previously explored actively to get food rewards at specific places. In contrast, in current experiments, these mice had never actively explored the space shown in the movie, nor obtained any rewards. Active exploration of a maze, combined with spatially localized rewards engages multisensory mechanisms resulting in increased place cell activation (*Mehta et al., 1997*; *Moore et al., 2021*; *Mehta and McNaughton, 1997*) which are entirely missing in these experiments during passive viewing of a movie, presented monocularly, without any other multisensory stimuli and without any rewards. Compared to their spontaneous activity, about half of CA1 and CA3 neurons shutdown during spatial exploration and this shutdown is even greater in the DG. Furthermore, compared to the exploration of a real-world maze, exploration of a visually identical virtual world causes 60% reduction in CA1 place cell activation (*Ravassard et al., 2013*). In contrast, there was no evidence of neural shutdown during the movie presentation compared to gray screen spontaneous epochs (*Figure 1—figure supplement 8*). Similarly, the number of place-fields (in CA1) per cell on a long track is positively correlated with the mean firing rate of the cell (*Rich et al., 2014*), which was not seen here for CA1 movie-fields.

A recent study showed that CA1 neurons encode the distance, angle, and movement direction of motion of a vertical bar of light (*Purandare et al., 2022*), consistent with the position of hippocampus in the visual circuitry (*Felleman and Van Essen, 1991*). Do those findings predict the movie tuning herein? There are indeed some similarities between the two experimental protocols – purely passive optical motion without any self-motion or rewards. However, there are significant differences too; similar to place cells in the real and virtual worlds (*Aghajan et al., 2015*), all the cells tuned to the moving bar of light had single receptive fields with elevated responses lasting a few seconds; there were neither punctate responses nor even 10-fold variation in neural field durations, let alone the 1000-fold change reported here. Finally, those results were reported only in area CA1, while the results presented here cover nearly all the major stations of the visual hierarchy.

Notably, hippocampal neurons did not encode Gabor patches or drifting gratings in the same dataset, indicating the importance of temporally continuous sequences of images for hippocampal activation (*Siegle et al., 2021*). This is consistent with the hypothesis that the hippocampus is involved in coding spatial sequences (*Mehta, 2015*; *Buzsáki and Tingley, 2018*; *Foster and Knierim, 2012*). However, unlike place cells that degrade in immobile rats, hippocampal movie tuning was unchanged in the immobile mouse. Furthermore, the scrambled sequence too was presented in the same sequence many times, yet movie tuning dropped to chance level in the hippocampal areas. Unlike visual areas, scrambled sequence response of hippocampal neurons could not be rearranged to obtain the continuous movie response. This shows the importance of continuous, episodic content instead of mere sequential recurrence of unrelated content for rodent hippocampal activation. We hypothesize that similar to place cells, movie-field responses without task demand would play a role, to be determined, in episodic memory. Further work involving a behavior report for the episodic content can potentially differentiate between the sequence coding described here and the contribution of episodically meaningful content. However, the nature of movie selectivity tested so far in humans was different (recall of famous, short movie clips [*Gelbard-Sagiv et al., 2008*], or at event boundaries [*Zheng et al., 2022*]) than in rodents here (human movie, selectivity to specific movie segments).

## Broader outlook

Our findings open up the possibility of studying thalamic, cortical, and hippocampal brain regions in a simple, passive, and purely visual experimental paradigm and extend comparable convolutional neural networks (*de Vries et al., 2020*) to have the hippocampus at the apex (*Felleman and Van Essen, 1991*). Furthermore, our results here bridge the long-standing gap between the hippocampal rodent and human studies (*Zheng et al., 2022*; *Rutishauser et al., 2006*; *Silson et al., 2021*; *King et al., 2021*), where natural movies can be decoded from fMRI (functional magnetic resonance imaging) signals in immobile humans (*Nishimoto et al., 2011*). This brain-wide mega-scale encoding of a human movie episode and enhanced preference for visual continuity in the hippocampus compared to visual areas supports the hypothesis that the rodent hippocampus is involved in non-spatial episodic memories, consistent with classic findings in humans (*Scoville and Milber, 1957*) and in agreement with a more generalized, representational framework (*Nadel and Peterson, 2013*; *Nadel and Hardt, 2011*) of episodic memory where it encodes temporal patterns. Similar responses are likely across

different species, including primates. Thus, movie-coding can provide a unified platform to investigate the neural mechanisms of episodic coding, learning, and memory.

## Methods

### Experiments

We used the Allen Brain Observatory – Neuropixels Visual Coding dataset (2019 Allen Institute, https://portal.brain-map.org/explore/circuits/visual-coding-neuropixels). This website and related publication (*Siegle et al., 2021*) contain detailed experimental protocol, neural recording techniques, spike sorting etc. Data from 24 mice (16 males, $n$ = 13 C57BL/6J wild-type, $n$ = 2 Pvalb-IRES-Cre×Ai32, $n$ = 6 Sst-IRES-Cre×Ai32, and $n$ = 3 Vip-IRES-Cre×Ai32) from the 'Functional connectivity' dataset were analyzed herein. Prior to implantation with Neuropixel probes, mice passively viewed the entire range of images including drifting gratings, Gabor patches and movies of interest here. Videos of the body and eye movements were obtained at 30 Hz and synced to the neural data and stimulus presentation using a photodiode. Movies were presented monocularly on an LCD monitor with a refresh rate of 60 Hz, positioned 15 cm away from the mouse's right eye and spanned 120° × 95°. Thirty trials of the continuous movie presentation were followed by 10 trials of the scrambled movie. Next was a presentation of drifting gratings, followed by a quiet period of 30 min where the screen was blank. Then the second block of drifting gratings, scrambled movie and continuous movie was presented. After surgery, all mice were single housed and maintained on a reverse 12 hr light cycle in a shared facility with room temperatures between 20 and 22°C and humidity between 30% and 70%. All experiments were performed during the dark cycle.

Neural spiking data were sampled at 30 kHz with a 500-Hz high pass filter. Spike sorting was automated using Kilosort2 (*Stringer et al., 2019*). Output of Kilosort2 was post-processed to remove noise units, characterized by unphysiological waveforms. Neuropixel probes were registered to a common co-ordinate framework (*Wang, 2020*). Each recorded unit was assigned to a recording channel corresponding to the maximum spike amplitude and then to the corresponding brain region. Broad spiking units identified as those with average spike waveform duration (peak to trough) between 0.45 and 1.5 ms and those with mean firing rates above 0.5 Hz were analyzed throughout, except *Figure 1—figure supplement 8*.

### Movie tuning quantification

The movie consisted of 900 frames: 30 s total, 30 Hz refresh rate, 33.3 ms per frame. At the first level of analysis, spike data were split into 900 bins, each 33.3 ms wide (the bin size was later varied systematically to detect mega-scale tuning, see below). The resulting tuning curves were smoothed with a Gaussian window of $\sigma$ = 66.6 ms or two frames. The degree of modulation and its significance was estimated by the sparsity $s$ as below, and as previously described (*Purandare et al., 2022*; *Ravassard et al., 2013*).

$$s = 1 - \frac{1}{N} \frac{\left(\sum_n r_n\right)^2}{\left(\sum_n r_n^2\right)}$$

where $r_n$ is the firing rate in the $n$th frame or bin and $N$ = 900 is the total number of bins. This is equivalent to 'lifetime sparseness', used previously (*de Vries et al., 2020*; *Vinje and Gallant, 2000*), except for the normalization factor of $(1 - 1/N)$, which is close to unity, when $N$ is close to 900 as in the case of movies. Statistical significance of sparsity was computed using a bootstrapping procedure, which does not assume a normal distribution. Briefly, for each cell, the spike train as a function of the frame number from each trial was circularly shifted by different amounts and the sparsity of the randomized data computed. This procedure was repeated 100 times with different amounts of random shifts. The mean value and standard deviation of the sparsity of randomized data were used to compute the z-scored sparsity of observed data using the function z-score in MATLAB. The observed sparsity was considered statistically significant if the z-scored sparsity of the observed spike train was greater 2, which corresponds to p < 0.023 in a one-tailed t-test. A similar method was used to quantify significance of the scrambled movie tuning, as well as for the subset of data with only stationary epochs, or its equivalent subsample (see below). Middle 20 trials of the continuous movie were used in

comparisons with the scrambled movie in *Figure 4*, to ensure a fair comparison by using same number of trials, with similar time delays across measurements.

In addition to sparsity, we quantified movie tuning using two other measures.

Depth of modulation = $(r_{max} - r_{min})/(r_{max} + r_{min})$, where $r_{max}$ and $r_{min}$ are the largest and lowest firing rates across movie frames, respectively.

Mutual information

$$\left(MI\right) = \sum_C p\left(C | frame_n . log_2 \frac{p\left(C \vee frame_n\right)}{p\left(C\right)}\right)$$

where

$$p\left(C\right) = \sum_n p\left(frame_n\right) . p\left(C | frame_n\right)$$

and $C$ is the average spike count in 0.033-s window which corresponds to 1 movie frame. $p\left(frame_n\right)$ is 1/900, as all frames were presented equal number of times. Statistical significance of these alternative measures of selectivity was computed similar to that for sparsity and is detailed in *Figure 1—figure supplement 3*.

## Stationary epoch and SWR-free epoch identification

To eliminate the confounding effects of changes in behavioral state associated with running, we repeated our analysis in stationary epochs, defined as epochs when the running speed remained less than 2 cm/s for this period, as well as for at least 5 s before and after this period. Analysis was further restricted to sessions with at least 5 total minutes of these epochs during the 60 trials of continuous movie presentation. To account for using lesser data of the stationary epochs, we compared the tuning using a random subsample of data, regardless of running or stopping and compared the two results for difference in selectivity.

Similarly, to remove epochs of SWRs, we first computed band passed power in the hippocampal (CA1) recording sites in the 150–250 Hz range. SWR occurrence was noted if any of the best five sites in CA1 (those with highest theta (5–12 Hz) to delta (1–4 Hz) ratio), or the median SWR across all CA1 sites exceeded their respective 3 standard deviations of power. To remove SWRs, we removed frames corresponding to ±0.5-s around the SWR occurrence and recomputed movie tuning in the remaining data. Similar to the stationary epoch calculation above, we compared tuning to an equivalent random subset to account for loss of data.

## Pupil dilation and theta power comparisons

To assess the contribution of arousal state on movie tuning, we re-calculated *z*-scored sparsity in epochs with high versus low pupil dilation. The pupil was tracked at a 30-Hz sampling rate, and the height and width of the elliptical fit as provided in the publicly available dataset was used. For each session, the pupil area thus calculated was split into two equal halves, by using data above and below the 50th percentile. The resultant *z*-scored sparsity is reported in *Figure 1—figure supplement 7*.

Similarly, the theta power computed from the band passed local field potential signal in the 5–12 Hz range was split into two equal data subsegments. The channel from CA1, with the highest average theta to delta (1–4 Hz) power ratio was nominated as the channel to be used for these calculations. Movie tuning in data with high and low theta power thus separated is reported in *Figure 1—figure supplement 7*.

## Mega-scale movie-field detection in tuned neurons

For neurons with significant movie-sparsity, i.e., movie tuned, the movie response was first recalculated at a higher resolution of 3.33 ms (10 times the frame rate of 33.3 ms). The *findpeaks* function in MATLAB was used to obtain peaks with *prominence* larger than 110% (1.1×) the range of firing variation obtained by chance, as determined from a sample shuffled response. This calculation was repeated at different smoothing values (logarithmically spaced in 10 Gaussian smoothing schemes with $\sigma$ ranging from 6.7 to 3430 ms), to ensure that long as well as short movie-fields were reliably detected and treated equally. For frames where overlapping peaks were found at different smoothing levels, we employed a comparative algorithm to only select the peak(s) with higher prominence score.

This score was obtained as the ratio of the peak's prominence to the range of fluctuations in the correspondingly smoothed shuffle. This procedure was conducted iteratively, in increasing order of smoothing. If a broad peak overlapped with multiple narrow ones, the sum of scores of the narrow ones was compared with the broad one. To ensure that peaks at the beginning as well as the end of the movie frames were reliably detected, we circularly wrapped the movie response, for the observed as well as shuffle data.

## Identifying frames with significant deviations in multiple single-unit activity

First, the average response across tuned neurons for each brain region was computed for each movie frame, after normalizing the response of each cell by the peak firing response. This average response was used as the observed 'Multiple single-unit activity (MSUA)' in *Figure 3*. To compute chance level, individual neuron responses were circularly shifted with respect to the movie frames to break the frame to firing rate association but maintain overall firing rate modulation. 100 such shuffles were used, and for each shuffle, the shuffled MSUA response was computed by averaging across neurons. Across these 100 shuffles, mean and standard deviation was obtained for all frames, and used to compute the z-score of the observed MSUA. To obtain significance at p = 0.025 level, Bonferroni correction was applied, and the appropriate *z*-score (4.04) level was chosen. The number of frames in the observed MSUA above (and below) this level is further quantified in *Figure 3*. The firing deviation for these frames was computed as the ratio between the mean observed MSUA and the mean shuffled MSUA, reported as a percentage, for frames corresponding to *z*-score greater than +4 or less than −4. To obtain a total firing rate report, where each spike gets equal vote, we computed the total firing response by computing the total rate across all tuned neurons (and averaging by the number of neurons) in *Figure 3* and across all neurons in *Figure 3—figure supplement 2*.

## Population vector overlap

To evaluate the properties of a population of cells, movie presentations were divided into alternate trials, yielding even and odd blocks (*Resnik et al., 2012*). Population vector overlap was computed between the movie responses calculated separately for these two blocks of trials. Population vector overlap between frames *x* of the even trials and frame *y* of the odd trials was defined as the Pearson correlation coefficient between the vectors ($R_{1,x}$, $R_{2,x}$, … $R_{N,x}$) and ($R_{1,y}$, $R_{2,y}$, … $R_{N,y}$), where $R_{n,x}$ is the mean firing rate response of the *n*th neuron to the *x*th movie frame. *N* is the total number of neurons used, for each brain region. This calculation was done for *x* and *y* ranging from 1 to 900, corresponding to the 900 movie frames. The same method was used for tuned and untuned neurons in continuous movie responses in *Figure 3—figure supplement 1*, and for scrambled sequence responses in *Figure 4—figure supplement 6*.

## Decoding analysis

Methods similar to those previously described were used (*Purandare et al., 2022*; *Wilson and McNaughton, 1993*). For tuned cells, the 60 trials of continuous movie were each decoded using all other trials. Mean firing rate responses in the 59 trials for 900 frames were used to compute a 'look-up' matrix. Each neuron's response was normalized between 0 and 1. At each frame in the 'observed' trial, the correlation coefficient was computed between the population vector response in this trial and the look-up matrix. The frame corresponding to the maximal correlation was denoted as the decoded frame. Decoding error was computed as the average of the absolute difference between actual and decoded frames, across the 900 frames of the movie. For comparison, shuffle data were generated by randomly shuffling the cell–cell pairing of the look-up matrix and 'observed response'. To enable a fair comparison of decoding accuracy across brain regions, the tuned cells from each brain region were subsampled, and a random selection of 150 cells was used. A similar procedure was used for the 20 trials of the scrambled sequence, and the corresponding middle 20 trials of the continuous movie were used here for comparison.

## Rearranged scrambled movie analysis

To differentiate the effects of visual content versus visual continuity between consecutive frames, we compared the responses of the same neuron to the continuous movie and the scrambled sequence.

In the scrambled movie, the same visual frames as the continuous movie were used, but they were shuffled in a pseudo random fashion. The same scrambled sequence was repeated for 20 trials. The neural response was first computed at each frame of the scrambled sequence, keeping the frames in the chronological order of presentation. Then the scrambled sequence of frames was rearranged to recreate the continuous movie and the corresponding neural responses computed. To address the latency between movie frame presentation and its evoked neural response, which can differ across brain regions and neurons, this calculation was repeated for rearranged scrambled sequences with variable delays between $\tau$ = −500 to +500 ms (i.e., −150 to +150 frames of 3.33 ms resolution, in steps of five frames or 16.6 ms). The correlation coefficient was computed between the continuous movie response and this variable delayed response at each delay as $r_{measured}(\tau)$ = corrcoef($R_{continuous}$, $R_{scramble-rearranged}(\tau)$). $R_{continuous}$ is the continuous movie response, obtained at 3.33-ms resolution and similarly, $R_{scramble-rearranged}$ corresponds to the scrambled response after rearrangement, at the latency $\tau$. The latency $\tau$ yielding the largest correlation between the continuous and rearranged scrambled movie was designated as the putative response latency for that neuron. This was used in *Figure 4—figure supplement 4*. The value of $r_{measured}(\tau_{max})$ was bootstrapped using 100 randomly generated frame reassignments, and this was used to z-score $r_{measured}(\tau_{max})$, with z-score >2 as criterion for significance. The resultant z-score is reported in *Figure 4—figure supplement 4*.

The latency $\tau$ was rounded off for use with 33 ms bins and used to rearrange actual as well as shuffled data to compute the strength of tuning for scrambled presentation. Z-scored sparsity was computed as described above. This was compared with the z-scored sparsity of continuous movie as well as the scrambled movie data, without the rearrangement, and shown in *Figure 4—figure supplement 5*.

## Code availability

All analyses were performed using custom-written code in MATLAB version R2020a. Codes written for analysis and visualization are available on GitHub, at https://github.com/cspurandare/ELife_MovieTuning (*Purandare, 2023a*, copy archived at *Purandare, 2023b*).

## Acknowledgements

We thank the Allen Brain Institute for provision of the dataset, Dr. Josh Siegle for help with the dataset, Dr. Krishna Choudhary for proof-reading of the text, and Dr. Massimo Scanziani for input and feedback. This work was supported by grants to MRM by the National Institutes of Health NIH 1U01MH115746.

## Additional information

### Funding

| Funder | Grant reference number | Author |
|---|---|---|
| National Institutes of Health | 1U01MH115746 | Mayank Mehta |

The funders had no role in study design, data collection and interpretation, or the decision to submit the work for publication.

### Author contributions

Chinmay Purandare, Conceptualization, Data curation, Formal analysis, Validation, Investigation, Visualization, Writing – original draft, Writing – review and editing; Mayank Mehta, Conceptualization, Resources, Supervision, Funding acquisition, Investigation, Methodology, Writing – original draft, Project administration, Writing – review and editing

### Author ORCIDs

Chinmay Purandare ![ORCID] https://orcid.org/0000-0001-9225-0186
Mayank Mehta ![ORCID] https://orcid.org/0000-0003-2005-2468

### Ethics

No human subjects involved.

Reviewer #1 (Public Review): https://doi.org/10.7554/eLife.85069.3.sa1
Reviewer #3 (Public Review): https://doi.org/10.7554/eLife.85069.3.sa2
Author Response https://doi.org/10.7554/eLife.85069.3.sa3

---

## Additional files

### Supplementary files

• MDAR checklist

### Data availability

All data are publicly available at the Allen Brain Observatory - Neuropixels Visual Coding dataset (2019 Allen Institute, https://portal.brain-map.org/explore/circuits/visual-coding-neuropixels).

The following previously published dataset was used:

| Author(s) | Year | Dataset title | Dataset URL | Database and Identifier |
|---|---|---|---|---|
| Siegle JH, Jia X, Durand S | 2020 | Neuropixel | https://registry.opendata.aws/allen-brain-observatory | Registry of Open Data on AWS, allen-brain-observatory |

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
