## [Editor Report · eLife assessment]

This manuscript analyzes large-scale Neuropixels recordings from visual areas and hippocampus of mice passively viewing repeated clips of a movie and reports that neurons respond with elevated firing activities to specific, continuous sequences of movie frames. The **important** results support a role of rodent hippocampal neurons in general episode encoding and advance understanding of visual information processing across different brain regions. The strength of evidence for the primary conclusion was found to be **convincing**.

---

## [Referee Report · Reviewer #1 (Public Review)]

Taking advantage of a publicly available dataset, neuronal responses in both the visual and hippocampal areas to passive presentation of a movie are analyzed in this manuscript. Since the visual responses have been described in a number of previous studies (e.g., see Refs. 11-13), the value of this manuscript lies mostly on the hippocampal responses, especially in the context of how hippocampal neurons encode episodic memories. Previous human studies show that hippocampal neurons display selective responses to short (5 s) video clips (e.g. see Gelbard-Sagiv et al, Science 322: 96-101, 2008). The hippocampal responses in head-fixed mice to a longer (30 s) movie as studied in this manuscript could potentially offer important evidence that the rodent hippocampus encodes visual episodes.

The analysis strategy is mostly well designed and executed. A number of factors and controls, including baseline firing, locomotion, frame-to-frame visual content variation, are carefully considered. The inclusion of neuronal responses to scrambled movie frames in the analysis is a powerful method to reveal the modulation of a key element in episodic events, temporal continuity, on the hippocampal activity. The properties of movie fields are comprehensively characterized in the manuscript.

Comments on latest version:

The new analysis on how behavioral states and hippocampal ripples impacted the tuning of movie fields makes the main finding substantially more convincing. Other relatively minor concerns on the methodology and interpretation are also improved. I do not have further concerns.

---

## [Referee Report · Reviewer #3 (Public Review)]

In their study, Purandare & Mehta analyze large-scale single unit recordings from the visual system (LGN, V1, extrastriate regions AM and PM) and hippocampal system (DG, CA3, CA1 and subiculum) while mice monocularly viewed repeats of a 30s movie clip. The data were part of a larger release of publicly available recordings from the Allen Brian Observatory. The authors found that cells in all regions exhibited tuning to specific segments of the movie (i.e. "movie fields") ranging in duration from 20ms to 20s. The largest fractions of movie-responsive cells were in visual regions, though analyses of scrambled movie frames indicated that visual neurons were driven more strongly by visual features of the movie images themselves. Cells in the hippocampal system, on the other hand, tended to exhibit fewer "movie fields", which on average were a few seconds in duration, but could range from >50ms to as long as 20s. Unlike the visual system "movie fields" in the hippocampal system disappeared when the frames of the movie were scrambled, indicating that the cells encoded more complex (episodic) content, rather than merely passively reading out visual input.

The paper is conceptually novel since it specifically aims to remove any behavioral or task engagement whatsoever in the head-fixed mice, a setup typically used as an open-loop control condition in virtual reality-based navigational or decision making tasks (e.g. Harvey et al., 2012). Because the study specifically addresses this aspect of encoding (i.e. exploring effects of pure visual content rather than something task-related), and because of the widespread use of video-based virtual reality paradigms in different sub-fields, the paper should be of interest to those studying visual processing as well as those studying visual and spatial coding in the hippocampal system.

Comments on latest version:

The revised manuscript by Purandare et al. has been improved with the inclusion of additional analyses and discussion, and the changes mainly satisfy the concerns raised in the initial version of the manuscript.

Regarding the methods, it was particularly helpful that the authors took measures to consider the impact of different states of arousal (pupil diameter), mobility, and SWRs on the expression and significance of movie field tuning, considering the lack of a task structure or behavioral report. Relatedly, the additional metrics applied (information rate and depth of movie field modulation) substantiate the results as based on z-scored sparsity. The explanation of lifetime sparseness as used here vs. in the work of de Vries et al. 2020 was also helpful.

The addition of more clearly tuned cells also helps the study feel more rooted in solid ground. For clarity, and consistency with the rest of the paper, it would be helpful to add the sparseness metrics above the newly added neural data in the Figure supplements.

The Discussion also contains elements that help balance both it and the paper as a whole. It draws a clearer distinction between the representation of visual scenes rather than encoding the contents of episodic memory, clarifying that hippocampal neurons were more likely doing the former than the latter. It is also appreciated that the authors added discussion acknowledging that the cortical processing did not quite follow an apparent hierarchical order.

As a last observation, though the authors assert in their rebuttal that analysis of the visual content encoded in the movie fields is beyond the scope of the study, this would add an interesting dimension to the work. Because, to my awareness, much less is known regarding how the visual and hippocampal systems in rodents encode visual information when the visual input is dynamic and chunked, as with movies. It would prove an interesting addition to the more extensive work on the processing of static visual scenes.

---

## [Author Response]

The following is the authors’ response to the original reviews.

**eLife assessment**
This manuscript analyzes large-scale Neuropixels recordings from visual areas and hippocampus of mice passively viewing repeated clips of a movie and reports that neurons respond with elevated firing activities to specific, continuous sequences of movie frames. The important results support a role of rodent hippocampal neurons in general episode encoding and advance understanding of visual information processing across different brain regions. The strength of evidence for the primary conclusion is solid, but some technical limitations of the study were identified that merit further analyses.

We thank the editors and reviews for the assessment and reviews. We have provided clarifications and updated the manuscripts to address the seeming technical limitations that are perhaps due to some misunderstanding, please see below. We provide additional results that isolate the contribution of pupil diameter, sharpwave ripple and theta power to show that movie tuning cannot be explained by these nonspecific effects. Nor are these mere time cells or some other internally generated patterns due to many differences highlighted below.

**Reviewer #1 (Public Review):**
Taking advantage of a publicly available dataset, neuronal responses in both the visual and hippocampal areas to passive presentation of a movie are analyzed in this manuscript. Since the visual responses have been described in a number of previous studies (e.g., see Refs. 11-13), the value of this manuscript lies mostly on the hippocampal responses, especially in the context of how hippocampal neurons encode episodic memories. Previous human studies show that hippocampal neurons display selective responses to short (5 s) video clips (e.g. see Gelbard-Sagiv et al, Science 322: 96-101, 2008). The hippocampal responses in head-fixed mice to a longer (30 s) movie as studied in this manuscript could potentially offer important evidence that the rodent hippocampus encodes visual episodes.

We have now included citations to Gelbard-Sagiv et al. Science 2008 paper and many other references too, thank you for pointing that out. There are major differences between that study and ours.

a. The movies used in previous study contained very familiar, famous people and famous events, and the experiment was about the patient’s ability to recall those famous movie episodes. In our case the mice had seen this movie clip only in two habituation sessions before.

b. They did not look at the fine structure of neural responses below half a second whereas we looked at the mega-scale representations from 30ms to 30s.

c. The movie clips in that study were in full color with audio, we used an isoluminant, black-and-white, silent movie clip.

d. Their movie clips contained humans and was observed by humans, whereas our study mice observed a movie clip with humans and no mice or other animals.

The analysis strategy is mostly well designed and executed. A number of factors and controls, including baseline firing, locomotion, frame-to-frame visual content variation, are carefully considered. The inclusion of neuronal responses to scrambled movie frames in the analysis is a powerful method to reveal the modulation of a key element in episodic events, temporal continuity, on the hippocampal activity. The properties of movie fields are comprehensively characterized in the manuscript.

Thank you.

Although the hippocampal movie fields appear to be weaker than the visual ones (Fig. 2g, Ext. Fig. 6b), the existence of consistent hippocampal responses to movie frames is supported by the data shown. Interestingly, in my opinion, a strong piece of evidence for this is a "negative" result presented in Ext. Fig. 13c, which shows higher than chance-level correlations in hippocampal responses to same scrambled frames between even and odd trials (and higher than correlations with neighboring scrambled frames). The conclusion that hippocampal movie fields depend on continuous movie frames, rather than a pure visual response to visual contents in individual frames, is supported to some degree by their changed properties after the frame scrambling (Fig. 4).

Yes, hippocampal selectivity is not entirely abolished with scrambled movie, as we show in several figures (Figure 4d,g and Figure 4- figure supplement 6), but it is greatly reduced, far more than that in the afferent visual cortices. The fraction of tuned cells for scrambled movies dropped to 4.5% in hippocampus, which is close to the chance level of 3%. In contrast, in visual areas selectivity was still above 80%.

Significant overlap between even and odd trials is to be expected for the tuned cells. Without a significant overlap, i.e. a stable representation, they will not be tuned. Despite this, the correlation between even and odd trials for the (only 4.5% of) tuned cells in the hippocampus was more than 2-fold smaller than (more than 80% of) cells in visual cortices. This strongly supports our hypothesis that unlike visual cortices, hippocampal subfields depended very strongly on the continuity of visual information. We have now clarified this in the main text.

However, there are two potential issues that could complicate this main conclusion.One issue is related to the effect of behavioral variation or brain state. First, although the authors show that the movie fields are still present during low-speed stationary periods, there is a large drop in the movie tuning score (Z), especially in the hippocampal areas, as shown in Ext. Fig. 3b (compared to Ext. Fig. 2d). This result suggests a potentially significant enhancement by active behavior.

There seems to be some misunderstanding here. There was no major reduction in movie tuning during immobility or active running. As we wrote in the manuscript, the drop in selectivity during purely immobile epochs is because of reduction in the amount of data, not reduction in selectivity per se. Specifically, as the amount data reduces, the statistical strength of tuning (z-scored sparsity) reduces. For example, if we split the total of 60 trials worth of data into two parts, the amount of data reduces to about half in each part, leading to a seeming reduction in selectivity in both halves. Figure 1-figure supplement 4c shows nearly identical tuning in all brain regions during immobility (red bars) and equivalent subsamples (yellow-orange) chosen randomly from the entire data, including mobility and immobility. We also show that the movie tuning persists in sessions with and without prolonged running behavior (Figure 1-figure supplement 7), as well as by splitting the data based on pupil dilation or theta power. Please see below for more details.

Second, a general, hard-to-tackle concern is that neuronal responses could be greatly affected by changes in arousal or brain state (including drowsy or occasional brief slow-wave sleep state) in head-fixed animals without a task. Without the analysis of pupil size or local field potentials (LFPs), the arousal states during the experiment are difficult to know.

In the revised manuscript we show that the behavioral state effects cannot explain movie tuning. Specifically:

a. We compared sessions in which the mouse was mostly immobile versus sessions in which the mouse was mostly running. Movie tuned cells were found in both these cases (Figure 1-figure supplement 7).

b. We detected and removed all data around sharp-wave ripples (SWR). Movie tuning was unchanged in the remaining data. (Figure 1-figure supplement 6).

c. As a further control, we quantified arousal by two standard metrics. First within a session, we split the data into two groups, segments with high theta power and segments with low theta power. Significant movie tuning persisted in both.

d. Finally, pupil dilation is another common method to estimate arousal, so data within a session were split into two parts: those with pupil dilation versus constriction. Movie tuning remained significant in both parts. See the new Figure 1-figure supplement 7.

Many example movie fields in the presented raw data (e.g., Fig. 1c, Ext. Fig. 4) are broad with low-quality tuning, which could be due to broad changes in brain states. This concern is especially important for hippocampal responses, since the hippocampus can enter an offline mode indicated by the occurrence of LFP sharp-wave ripples (SWRs) while animals simply stay immobile. It is believed that the ripple-associated hippocampal activity is driven mainly by internal processing, not a direct response to external input (e.g., Foster and Wilson, Nature 440: 680, 2006). The "actual" hippocampal movie fields during a true active hippocampal network state, after the removal of SWR time periods, could have different quantifications that impact the main conclusion in the manuscript.

We included the broadly tuned hippocampal neurons to demonstrate the movie-field broadening compared to those in visual areas. We now include more examples with sharp movie fields in the hippocampal regions (Figure 1a-d right column, 2d and h, Figure 1-figure supplement 5 and Figure 2-figure supplement 1). Further, as stated above, we detected sharp-wave ripples and removed one second of data around SWR. Movie tuning was unchanged in the remaining data. Thus, movie tuning is not generated internally via SWR (Figure 1-figure supplement 6). See also Figure 1-figure supplement 7 and Figure 2-figure supplement 8 and the response above.

Another issue is related to the relative contribution of direct visual response versus the response to temporal continuity in movie fields. First, the data in Ext. Fig. 8 show that rapid frame-to-frame changes in visual contents contribute largely to hippocampal movie fields (similarly to visual movie fields).

There seems to be some misunderstanding here. That figure showed that the frame-to-frame changes in the visual content had the highest effect on visual areas MSUA and much weaker in hippocampus (Extended Data Fig. 8, as per previous version, now Figure3-figure supplement 2). For example, the depth of modulation (max – min) / (max + min) for MSUA was 21% and 24% for V1 but below 6% for hippocampal regions. Similarly, the MSUA was more strongly (negatively) correlated with F2F correlation for visual areas (r=0.48 to 0.56) than hippocampal (0.07 to 0.3). Similarly, comparing the number of peaks or their median widths, visual regions showed stronger correlation with F2F, and largest depth of modulation than hippocampal regions, barring handful exceptions (like CA3 correlation between F2F and median peak duration). This strongly supports our claim that visual regions generated far greater response of the frame-to-frame changes in the movie than hippocampal regions.

Interestingly, the data show that movie-field responses are correlated across all brain areas including the hippocampal ones.

In Figure 3c we compared the MSUA responses with normalization between brain regions. Amongst the 21 possible brain region pairs, 5 were uncorrelated, 7 were significantly negatively correlated and 9 were significantly positively correlated.

The changes in population overlap, number and widths of peaks are strongly correlated only between visual areas and some of the hippocampal region pairs. The correlation is much weaker for hippocampal-visual area pairs, but often significantly different from chance. This is quantified explicitly in the revised text Figure 3-figure supplement 2 with an additional correlation matrix at the right.

This could be due to heightened behavioral arousal caused by the changing frames as mentioned above, or due to enhanced neuronal responses to visual transients, which supports a component of direct visual response in hippocampal movie fields.

As shown in Figure 1-figure supplements 4,5,6 and 7 and described above, the effect of arousal as quantified by theta power of pupil diameter (or by accounting for running behavior or SWR occurrences) cannot explain the results in hippocampal areas and the correlations in multiunit responses are unrelated across many brain areas.

Second, the data in Ext. Fig. 13c show a significant correlation in hippocampal responses to same scrambled frames between even and odd trials, which also suggests a significant component of direct visual response.

This is plausible. The fraction of hippocampal cells which were significantly tuned for the scrambled presentation (4.5%) was close to chance level (3%), and this small subset of cells was used to compute the population overlap between even and odd trials in Figure 4-figure supplement 6 (Ext Fig. 13 with old numbering). As described above, this significant but small amount of tuning could generate significant population overlap, which is to be expected by construction.

Is there a significant component purely due to the temporal continuity of movie frames in hippocampal movie fields? To support that this is indeed the case, the authors have presented data that hippocampal movie fields largely disappear after movie frames are scrambled. However, this could be caused by the movie-field detection method (it is unclear whether single-frame field could be detected).

As described in the methods section, the movie-field detection algorithm had a resolution of 3.3ms resolution, which ensured that we could detect single frame fields. As reported, we did find such short movie fields in several cells in the visual areas. The sparsity metric used is agnostic to the ordering of the responses, and hence single frame field, and the resultant significant movie-tuning, if present, can be detected by our methods.

Another concern in the analysis is that movie-fields are not analyzed on re-arranged neural responses to scrambled movie frames. The raw data in Fig. 4e seem quite convincing. Unfortunately, the quantifications of movie fields in this case are not compared to those with the original movie.

We saw very few (3.6-4.9%) cells with significant movie tuning for scrambled presentation in the hippocampus. Hence, we did not quantify this earlier. This is now provided in new Figure 4-figure supplement 5. The amount of movie tuning for the scrambled presentation taken as-is, or after rearranging the frames is below 5% for all hippocampal brain regions and not significantly different between the two.

**Reviewer #2 (Public Review):**
Purandare and Mehta investigated the neural activities modulated by continuous and sequential visual stimuli composed of natural images, termed "movie-tuning," measured along the visuo-hippocampal network when the animals passively viewed a movie without any task demand. Neurons selectively responded to some specific parts of the movie, and their activity timescales ranged from tens of milliseconds to seconds and tiled the entire movie with their movie-fields. The movie-tuning was lost in the hippocampus but not in the visual cortices when the image frames were temporally scrambled, implying that the rodent hippocampus encoded the specific sequence of images.The authors have concluded that the neurons in the thalamo-cortical visual areas and the hippocampus commonly encode continuous visual stimuli with their firing fields spanning the mega-scale, but they respond to different aspects of the visual stimuli (i.e., visual contents of the image versus a sequence of the images). The conclusion of the study is fairly supported by the data, but some remaining concerns should be addressed.1. Care should be taken in interpreting the results since the animal's behavior was not controlled during the physiological recording.

This was done intentionally since plenty of research shows that task demand (e.g., Aronov and Tank, Nature 2017) can not only modulate hippocampal responses but also dramatically alter them. We have now provided additional figures (Figure 1-figure supplement 6 and 7) where we quantified the effects of the behavioral states (sharp wave ripples, theta power and pupil diameter), as well as the effect of locomotion (Figure 1-figure supplement 4). Movie tuning remained unaffected with these manipulations. Thus, movie tuning cannot be attributed to behavioral effects.

It has been reported that some hippocampal neuronal activities are modulated by locomotion, which may still contribute to some of the results in the current study. Although the authors claimed that the animal's locomotion did not influence the movie-tuning by showing the unaltered proportion of movie-tuned cells with stationary epochs only, the effects of locomotion should be tested in a more specific way (e.g., comparing changes in the strength of movie-tuning under certain locomotion conditions at the single-cell level).

Single cell analysis of the effect of locomotion and visual stimulation is underway, and beyond the scope of the current work. As detailed in Figure 1-figure supplement 4, we have ensured that in spite of the removal of running or stationary epochs, as well as removal of sharp wave ripple events (Figure 1-figure supplement 6) movie tuning persists. Further, we now provide examples of strongly tuned cells from sessions with predominantly running or predominantly stationary behavior (Figure 1-figure supplement 7).

1. The mega-scale spanning of movie-fields needs to be further examined with a more controlled stimulus for reasonable comparison with the traditional place fields. This is because the movie used in the current study consists of a fast-changing first half and a slow-changing second half, and such varying and ununified composition of the movie might have largely affected the formation of movie-fields. According to Fig. 3, the mega-scale spanning appears to be driven by the changes in frame-to-frame correlation within the movie. That is, visual stimuli changing quickly induced several short fields while persisting stimuli with fewer changes elongated the fields.

Please note that a strong correlation between the speed at which the movie scene changed across frames was correlated with movie-field width in the visual areas, but that correlation was much weaker in the hippocampal areas correlation values - (LGN +0.61, V1 +0.51, AM-PM +0.55 vs. DG +0.39, CA3 +0.58, CA1 +0.42, SUB +0.24). Please see Figure 3-figure supplement 2 and the quantification of correlation between frame-to-frame changes in the movie and the properties of movie fields.

The presentation of persisting visual input for a long time is thought to be similar to staying in one place for a long time, and the hippocampal activities have been reported to manifest in different ways between running and standing still (i.e., theta-modulated vs. sharp wave ripple-based). Therefore, it should be further examined whether the broad movie-fields are broadly tuned to the continuous visual inputs or caused by other brain states.

As shown in Figure 1-figure supplement 6, movie field properties are largely unchanged when SWR are removed from the data, or when the effect of pupil diameter or theta power were factored for (Figure 1-figure supplement 7).

1. The population activities of the hippocampal movie-tuned cells in Fig. 3a-b look like those of time cells, tiling the movie playback period. It needs to be clarified whether the hippocampal cells are actively coding the visual inputs or just filling the duration.

Tiling patterns would be observed when the maxima are sorted in any data, even for random numbers. This alone does not make them time cells. The following observations suggest that movie fields cannot be explained as being time cells.

a. Time cells mostly cluster at the beginning of a running epoch (Pastalkova et al. Science 2008, MacDonald et al. Neuron 2011) and they taper off towards the end. Such large clustering is not visible in these tiling plots for movie tuned cells.

b. Time fields become wider as the temporal duration progresses (Pastalkova et al. Science 2008, MacDonald et al. Neuron 2011) as the encoded temporal duration increases. This is not evident in any movie fields.

c. Widths of movie fields in visual areas, and to a smaller extent in the hippocampal areas, were clearly modulated by the visual content, like the change from one frame to the next (F2F correlation, Figure 3-figure supplement 2).

d. Tiling pattern of movie fields was found in visual areas too, with qualitatively similar pattern as hippocampus. Clearly, visual area responses are not time cells, as shown by the scrambled stimulus experiment. Here, neural selectivity could be recovered by rearranging them based on the visual content of the continuous movie, and not the passage of time.

The scrambled condition in which the sequence of the images was randomly permutated made the hippocampal neurons totally lose their selective responses, failing to reconstruct the neural responses to the original sequence by rearrangement of the scrambled sequence. This result indirectly addressed that the substantial portion of the hippocampal cells did not just fill the duration but represented the contents and temporal order of the images. However, it should be directly confirmed whether the tiling pattern disappeared with the population activities in the scrambled condition (as shown in Extended Data Fig. 11, but data were not shown for the hippocampus).

As stated above for the continuous movie, tiling pattern alone does not mean those are time cells. Further, tuning, and tiling pattern remained intact with scrambled movie in the visual cortices but not in hippocampus. We now added a new supplement figure – Figure 4-figure supplement 5 where we compared the movie tuning for scrambled presentation with and without rearranging the frames. Hippocampal tuning remains at chance levels.

**Reviewer #3 (Public Review):**
In their study, Purandare & Mehta analyze large-scale single unit recordings from the visual system (LGN, V1, extrastriate regions AM and PM) and hippocampal system (DG, CA3, CA1 and subiculum) while mice monocularly viewed repeats of a 30s movie clip. The data were part of a larger release of publicly available recordings from the Allen Brian Observatory. The authors found that cells in all regions exhibited tuning to specific segments of the movie (i.e. "movie fields") ranging in duration from 20ms to 20s. The largest fractions of movie-responsive cells were in visual regions, though analyses of scrambled movie frames indicated that visual neurons were driven more strongly by visual features of the movie images themselves. Cells in the hippocampal system, on the other hand, tended to exhibit fewer "movie fields", which on average were a few seconds in duration, but could range from >50ms to as long as 20s. Unlike the visual system "movie fields" in the hippocampal system disappeared when the frames of the movie were scrambled, indicating that the cells encoded more complex (episodic) content, rather than merely passively reading out visual input.The paper is conceptually novel since it specifically aims to remove any behavioral or task engagement whatsoever in the head-fixed mice, a setup typically used as an open-loop control condition in virtual reality-based navigational or decision making tasks (e.g. Harvey et al., 2012). Because the study specifically addresses this aspect of encoding (i.e. exploring effects of pure visual content rather than something task-related), and because of the widespread use of video-based virtual reality paradigms in different sub-fields, the paper should be of interest to those studying visual processing as well as those studying visual and spatial coding in the hippocampal system. However, the task-free approach of the experiments (including closely controlling for movement-related effects) presents a Catch-22, since there is no way that the animal subjects can report actually recognizing or remembering any of the visual content we are to believe they do.

Our claim is that these are movie scene evoked responses. We make no claims about the animal’s ability to recognize or remember the movie content. That would require entirely different set of experiments. Meanwhile, we have shown that these results are not an artifact of brain states such as sharp wave ripples, theta power or pupil diameter (Figure1-figure supplement 6 and 7) or running behavior (Figure 1-figure supplement 4). Please see above for a detailed response.

We must rely on above-chance-level decoding of movie segments, and the requirement that the movie is played in order rather than scrambled, to indicate that the hippocampal system encodes episodic content of the movie. So the study represents an interesting conceptual advance, and the analyses appear solid and support the conclusion, but there are methodological limitations.

It is important to emphasize that these responses could constitute episodic responses but does not prove episodic memory, just as place cell responses constitute spatial responses but that does not prove spatial memory. The link between place cells and place memory is not entirely clear. For example, mice lacking NMDA receptors have intact place cells, but are impaired in spatial memory task (McHugh et al. Cell 1996), whereas spatial tuning was virtually destroyed in mice lacking GluR1 receptors, but they could still do various spatial memory tasks (Resnik et al. J. Neuro 2012).

The experiments about episodic memory would require an entirely different set of experiments that involve task demand and behavioral response, which in turn would modify hippocampal responses substantially, as shown by many studies. Our hypothesis here, is that just like place cells, these episodic responses without task demand would play a role, to be determined, in episodic memory. We have emphasized this point in the main text (Ln 391-393 in the revised manuscript).

Major concerns:1. A lot hinges on hinges on the cells having a z-scored sparsity >2, the cutoff for a cell to be counted as significantly modulated by the movie. What is the justification of this criterion?

The z-scored sparsity (z>2) corresponds to p<0.03. This would mean that 3% of the results could appear by chance. Hence, z>2 is a standard method used in many publications. Another advantage of z-scored sparsity is that it is relatively insensitive to the number of spikes generated by a neuron (i.e. the mean firing rate of the neuron and the duration of the experiment). In contrast, sparsity is strongly dependent on the number of spikes which makes it difficult to compare across neurons, brain regions and conditions (See Supplement S5 Acharya et al. Cell 2016).

To further address this point, we compared our z-scored sparsity measure with 2 other commonly used metrics to quantify neural selectivity, depth of modulation and mutual information (Figure 1-figure supplement 3). Comparable movie tuning was obtained from all 3 metrics, upon z-scoring in an identical fashion.

It should be stated in the Results. Relatedly, it appears the formula used for calculating sparseness in the present study is not the same as that used to calculate lifetime sparseness in de Vries et al. 2020 quoted in the results (see the formula in the Methods of the de Vries 2020 paper immediately under the sentence: "Lifetime sparseness was computed using the definition in Vinje and Gallant").

The definition of sparsity we used is used commonly by most hippocampal scientists (Treves and Rolls 1991, Skaggs et al. 1996, Ravassard et al. 2013). Lifetime sparseness equation used by de Vries et al. 2020, differs from us by just one constant factor (1-1/N) where N=900 is the number of frames in the movie. This constant factor equals (1-1/900)=0.999. Hence, there is no difference between the sparsity obtained by these two methods. Further, z-scored sparsity is entirely unaffected by such constant factors. We have clarified this in the methods of the revised manuscript.

To rule out systematic differences between studies beyond differences in neural sampling (single units vs. calcium imaging), it would be nice to see whether calculating lifetime sparseness per de Vries et al. changed the fraction "movie" cells in the visual and hippocampal systems.

As stated above, the two definitions of sparsity are virtually identical and we obtained similar results using two other commonly used metrics, which are detailed in Figure 1-figure supplement 3.

1. In Figures 1, 2 and the supplementary figures-the sparseness scores should be reported along with the raw data for each cell, so the readers can be apprised of what types of firing selectivity are associated with which sparseness scores-as would be shown for metrics like gridness or Raleigh vector lengths for head direction cells. It would be helpful to include this wherever there are plots showing spike rasters arranged by frame number & the trial-averaged mean rate.

As shown in several papers (Aghajan et al Nature Neuroscience 2015, Acharya et al., Cell 2016) raw sparsity (or information content) are strongly dependent on the number of spikes of a neuron. This makes the raw values of these numbers impossible to compare across cells, brain regions and conditions. (Please see Supplement S5 from Acharya et al., Cell 2016 for details). Including the data of sparsity would thus cause undue confusion. Hence, we provide z-scored sparsity. This metric is comparable across cells and brain regions, and now provided above each example cell in Figure 1 and Figure 1-figure supplement 2.

1. The examples shown on the right in Figures 1b and c are not especially compelling examples of movie-specific tuning; it would be helpful in making the case for "movie" cells if cleaner / more robust cells are shown (like the examples on the left in 1b and c).

We did not put the most strongly tuned hippocampal neurons in the main figures so that these cells are representative of the ensemble and not the best possible ones, so as to include examples with broad tuning responses. We have clarified in the legend that these cells are some of the best tuned cells. Although not the cleanest looking, the z-scored sparsity mentioned above the panels now indicates how strongly they are modulated compared to chance levels. Additional examples, including those with sharply tuned responses are shown in Figure 1-figure supplement 5 and Figure 2-figure supplement 1.

1. The scrambled movie condition is an essential control which, along with the stability checks in Supplementary Figure 7, provide the most persuasive evidence that the movie fields reflect more than a passive readout of visual images on a screen. However, in reference to Figure 4c, can the authors offer an explanation as to why V1 is substantially less affected by the movie scrambling than it's main input (LGN) and the cortical areas immediately downstream of it? This seems to defy the interpretation that "movie coding" follows the visual processing hierarchy.

This is an important point, one that we find very surprising as well. Perhaps this is related to other surprising observations in our manuscript, such as more neurons appeared to be tuned to the movie than the classic stimuli. A direct comparison between movie responses versus fixed images is not possible at this point due to several additional differences such as the duration of image presentations and their temporal history.

The latency required to rearrange the scrambled responses (60ms for LGN, 74ms for V1, 91ms for AM/PM) supports the anatomical hierarchy. The pattern of movie tuning properties was also broadly consistent between V1 and AM/PM (Figure 2).

However, all metrics of movie selectivity (Figure 2) to the continuous movie showed a consistent pattern that was the exact opposite pattern of the simple anatomical hierarchy: V1 had stronger movie tuning, higher number of movie fields per cell, narrower movie-field widths, larger mega-scale structure, and better decoding than LGN. V1 was also more robust to the scrambled sequence than LGN. One possible explanation is that there are other sources of inputs to V1, beyond LGN, that contribute significantly to movie tuning. This is an important insight and we have modified the discussion (Ln 315-325) to highlight this.

Relatedly, the hippocampal data do not quite fit with visual hierarchical ordering either, with CA3 being less sensitive to scrambling than DG. Since the data (especially in V1) seem to defy hierarchical visual processing, why not drop that interpretation? It is not particularly convincing as is.

The anatomical organization is well established and an important factor. Even when observations do not fit the anatomical hierarchy, it provides important insights about the mechanisms. All properties of movie tuning (Figure 2) –the strength of tuning, number of movie peaks, their width and decoding accuracy firmly put visual areas upstream of hippocampal regions. But, just like visual cortex there are consistent patterns that do not support a simple feed-forward anatomical hierarchy. We have pointed out these patterns so that future work can build upon it.

1. In the Discussion, the authors argue that the mice encode episodic content from the movie clip as a human or monkey would. This is supported by the (crucial) data from the scrambled movie condition, but is nevertheless difficult to prove empirically since the animals cannot give a behavioral report of recognition and, without some kind of reinforcement, why should a segment from a movie mean anything to a head-fixed, passively viewing mouse?

We emphasize once again that our claim is about the nature of encoding of the movie across these neurons. We make no claims about whether this forms a memory or whether the mouse is able to recognize the content or remember it. Despite decades of research, similar claims are difficult to prove for place cells, with plenty of counter examples (See the points above). The important point here is that despite any cognitive component, we see remarkably tuned responses in these brain areas. Their role in cognition would take a lot more effort and is beyond the scope of the current work.

Would the authors also argue that hippocampal cells would exhibit "song" fields if segments of a radio song-equally arbitrary for a mouse-were presented repeatedly? (reminiscent of the study by Aronov et al. 2017, but if sound were presented outside the context of a task). How can one distinguish between mere sequence coding vs. encoding of episodically meaningful content? One or a few sentences on this should be added in the Discussion.

Aronov et al 2017, found the encoding of an audio sweep in hippocampus when the animals were doing a task (release the lever at a specific frequency to obtain a reward). However, without a task demand they found that hippocampal neurons did not encode the audio sequence beyond chance levels. This is at odds with our findings with the movie where we see strong tuning despite any task demand or reward. These results are consistent with but go far beyond our recent findings that hippocampal (CA1) neurons can encode the position and direction of motion of a revolving bar of light (Purandare et al. Nature 2022). Please see Ln 373-382 for related discussion.

These responses are unlikely to be mere sequence responses since the scrambled sequence was also fixed sequence that was presented many times and it elicited reliable responses in visual areas, but not in hippocampus. Hence, we hypothesize that hippocampal areas encode temporally related information, i.e. episodic content. We have modified the discussion to address these points.

**Reviewer #1 (Recommendations For The Authors):**
1. Are LFP data available in the data set? If so, can SWRs identified and removed to refine the quantification of movie fields?

Done, see Figure 1-figure supplement 6.

1. Can movie fields be analyzed in re-arranged neural responses (Fig. 4e) and compared to those in other cases already shown (Fig. 4b, c)?

Done, even after rearrangement the strength of movie tuning for the scrambled presentation was low, and below 5% in all hippocampal regions. See Figure 4-figure supplement 5 for details.

1. It seems the authors are not fully committed to a main conclusion in the present manuscript. The title and abstract seem to emphasize the similar movie responses across visual and hippocampal areas, but the introduction and discussion emphasize the episode encoding of hippocampal neurons. The writing could be more consistent and the main message could be clearer.

Selective responses to the continuous movie showed similar patterns (prevalence of tuning, multi-peaked nature, relation with frame to frame changes in visual images) between visual and hippocampal regions. But the visual responses to scrambled presentation could be rearranged, and the latency for rearrangement increased from LGN to V1 to AM-PM. On the other hand, selectivity to the scrambled presentation was virtually abolished in hippocampus, and responses could not be rearranged to resemble the continuous movie sequences. To reconcile these differences, we have hypothesized here that the hippocampal responses are episodic in nature, and rely on temporal continuity, whereas the visual regions rely directly on the visual content in the images.

1. Line #158: "Net movie-field discharges was also comparable across brain areas...". This statement is not supported by Fig. 2g, which shows a wide range of median values across brain areas.

Thank you for pointing this out. The normalized firing in movie-fields used in that figure are within 3x between V1 and subiculum. We have modified the text to contrast this with the 10x difference between movie-field durations.

1. Line #253: What the two numbers (87.8%, 10.6%) mean is unclear (mean or median values). These numbers also appear inconsistent with the mean+-se values in Fig. 4 legend.

The numbers mentioned on Ln253, in the main text reflect the median visual continuity index, combining across cells from hippocampal or visual regions. On the other hand, values reported in the Fig 4 legend are for V1 and subiculum, which are the regions with smallest and largest visual continuity index, respectively. We have re-written the main text, and legends for better clarity.

1. The Gelbard-Sagiv et al paper (Science 322: 96-101, 2008) could be cited and its relevance to the present study could be discussed.

Done

1. Are there neurons recorded from a non-visual sensory or motor cortical area in the same experiment? This may provide a key negative control for the non-specific modulation caused by behavioral states or visual transients.

Owing to the nature of the experiments where the Allen Institute intended to study visual processing, we could not find any of the recorded brain regions without movie selectivity.

1. The differences in hippocampal and visual move fields between active and stationary time periods could be explicitly quantified.

We have shown several raster plots where the responses are quite similar during immobile and moving epochs. Our goal is to show that there is indeed comparable movie tuning when the animals is immobile versus any random state. Doing specific analysis of behavioral dependency is difficult because in many sessions the amount of time the mice ran in many sessions was very little. A thorough analysis overcoming these, and other challenges is beyond the scope of this paper.

**Reviewer #2 (Recommendations For The Authors):**
1. The methods to determine the boundaries of the movie-fields should be clarified, and the detected peaks and boundaries should be indicated in the relevant figures (e.g., Fig. 2c, 2d, and 2h) to help readers clearly understand how the movie-fields were defined and how the shapes of the movie-fields look like.

Done.

1. When testing the influence of locomotion on movie-tuning in Extended Data Fig. 3, a single cell-based analysis is further needed. For example, you need to check whether the z-scored sparsity within one cell varies or not depending on locomotion conditions (as in Extended Data Fig. 10a-c). In addition, it is recommended to exclude the cells significantly modulated by locomotion (e.g., running velocity) before defining the movie-tuned cells.

We now show example cells from sessions with or without prolonged running bouts in Figure 1-figure supplement 7 that have strong movie selectivity. We have also assessed the effects of theta power and pupil dilation on movie tuning in that figure. A more thorough analysis of the combined effects of locomotion and movie tuning is underway, but beyond the scope of the current work.

1. Regarding the time-cell-related issue raised in the public review, it would be nice if the authors confirm whether the tiling patterns of hippocampal subregions have been weakened by presenting the population activities for the scrambled condition as in the visual cortices in Extended Data Fig. 11a.

We have clarified in the earlier responses, please see above.

1. In Fig. 4 and Extended Data Fig. 3, the proportion of movie-tuned cells in the hippocampus seems to drop significantly after only a portion of trials under specific conditions were extracted. Although the authors addressed the stability issue by comparing the neural responses between even and odd trials, the concern about whether the movie-tuning is driven by a certain portion of trials still remains. To avoid such misunderstanding, as mentioned in comment no.2, tracking the changes in the z-scored sparsity of one cell between continuous and scrambled conditions should be provided. In addition, according to the methods, the scrambled condition was divided into two blocks of 10 trials each, possibly causing premature movie-tuned activities. Thus, it should be more appropriate to compare with the first 10 trials of each block in the continuous condition.

Done.

1. Explanations related to statistical analysis should be added to the methods sections. In Fig. 2a (and related figures with similar analysis), when comparing three or more groups, the Kruskal-Wallis test should be performed first to check whether there is a difference between the groups, and then pairwise comparisons should follow with adjusted p-values for multiple comparisons. Also, in Fig. 4b (and related figures), it seems that the K-S test was performed to test the changes in cell proportion by combining all brain regions, as far as I understand. However, it would be more appropriate to test the proportional changes by a Chi-square test within each region since the total numbers of cells should differ across the regions.

Yes, we have used the KS test throughout the analyses, unless otherwise mentioned or appropriate.

1. The labeling for firing rate is 'FR (sp/sec)' in Fig. 1, 2, and 4, but it is 'Firing rate (Hz)' in Fig. 3.

This has been fixed now, and only Firing rate (Hz), is used throughout. Thank you for pointing this out.

1. There is a typo in Extended Data Fig. 11b. "... across all tuned responses from (b)." It should be (a) instead of (b).

Done

**Reviewer #3 (Recommendations For The Authors):**
While the study presents an interesting dataset and conceptual approach, there are ways in which the manuscript should be strengthened.Minor concerns:1. Related to point (5) above, what content did the hippocampal "movie fields" encode? It would add a substantive dimension to the paper if the authors included examples of what segments of the movie the cells responded to. Are there "pan left" cells, or "man gets in the car" cells? Or was it more arbitrary than that? What is an example of a movie feature lasting 50ms that is stably encoded by a mouse hippocampal neuron?

We show example cells with very sharply tuned neural responses (Figure 2h). A thorough analysis of the visual content is in progress but beyond the scope of this paper.

1. Line 24-seems like it should read "Consistent presentation of the movie..." , with "ly" dropped from "consistent".

Done

1. Line 43-seems to be missing the article "a", and should read "...despite strong evidence for A hippocampal role in...".

We rewrote this sentence for better clarity

1. Line 54-to clarify, the higher visual areas recorded were the anteromedial (AM) and posterior-medial (PM) areas? The text additionally indicates a "medio-lateral" extrastriate area, but there is no such area. Can the text be revised to clear this up?

Sorry about this confusion, indeed we meant posterior-medial (PM). Thank you for pointing this out.

1. Line 84, "rate" should be pluralized to "rates".

Done

1. Line 108- the extra "But" at the start of the sentence should be removed.

Done

1. Figure 2h-was there any particular arrangement for the cells in this sub-panel? If not, could they be grouped by sub-region (or proximity between sub-regions) so it appears less arbitrary?

Done

1. Extended data 2 figure legend for (b) is missing a "that": "Fraction of selective neurons that was significantly above chance.... Ranging from 7.1% in CA

Done

1. Line 144-145, there is an extra "and" in the sentence: ".... were typically neither as narrow AND nor as prominent...."

Done

1. Line 203-the first word in the line should be "frames" (plural).

Done, thank you for pointing this out

1. Line 281-in "...scrambled sequence"-"sequence" should be plural. It looks like the same is true in line 882, in the legend title for Extended Data Fig. 11.

Since we only showed one scrambled sequence (which was repeated 20 times), we rewrote the relevant lines to be “the scrambled sequence”

1. Line 923-the first sentence of the legend for Extended Data Fig. 14-to what data or study are the authors referring to in saying that "More than 50% of hippocampal place cells shut down during maze exploration."? This was confusing, please clarify.

This reference has now been added.